# Oxidation of low-molecular weight organic compounds in cloud droplets: development of the JAMOC chemical mechanism in CAABA/MECCA (version 4.5.0)

Simon Rosanka[1], Rolf Sander[2], Andreas Wahner[1], and Domenico Taraborrelli[1]

[1]Forschungszentrum Jülich GmbH, Institute of Energy and Climate Research, IEK-8: Troposphere, Jülich, Germany
[2]Atmospheric Chemistry Department, Max-Planck Institute for Chemistry, Mainz, Germany

**Correspondence:** Simon Rosanka (s.rosanka@fz-juelich.de)

**Abstract.** The Jülich Aqueous-phase Mechanism of Organic Chemistry (JAMOC) is developed and implemented in the Module Efficiently Calculating the Chemistry of the Atmosphere (MECCA, version 4.5.0). JAMOC is an explicit in-cloud oxidation scheme for oxygenated volatile organic compounds (OVOCs), suitable for global model applications. It is based on a subset of the comprehensive CLoud Explicit Physico-chemical Scheme (CLEPS, version 1.0). The phase transfer of species containing up to ten carbon atoms is included, and a selection of species containing up to four carbon atoms reacts in the aqueous-phase. In addition, the following main advances are implemented: (1) simulating hydration and dehydration explicitly, (2) taking oligomerisation of formaldehyde, glyoxal and methylglyoxal into account, (3) adding further photolysis reactions, and (4) considering gas-phase oxidation of new outgassed species. The implementation of JAMOC in MECCA makes a detailed in-cloud OVOC oxidation model readily available for box as well as for regional and global simulations that are affordable with modern supercomputing facilities. The new mechanism is tested inside the box-model Chemistry As A Boxmodel Application (CAABA), yielding reduced gas-phase concentrations of most oxidants and OVOCs except for the nitrogen oxides.

## 1 Introduction

Aqueous-phase chemistry in cloud droplets differs significantly from gas-phase chemistry, mainly due to enhanced photolysis based on scattering effects within cloud droplets (Bott and Zdunkowski, 1987; Mayer and Madronich, 2004), faster reaction rates, and ion reactions that do not occur in the gas-phase (Herrmann, 2003; Epstein and Nizkorodov, 2012). Moreover, nitrogen monoxide (NO) to nitrogen dioxide ($NO_2$) conversion by peroxy radicals ($RO_2$) essentially does not take place in aqueous droplets because NO is insoluble. In the gas-phase, oxygenated volatile organic compounds (OVOCs) are mainly oxidised during daytime by hydroxyl radical (OH) and nitrate radical ($NO_3$) during nighttime (Herrmann et al., 2015). Even though ozone ($O_3$) is not very soluble, it can be taken up into cloud droplets where it is destroyed by:

$$O_3 + O_2^- \rightarrow O_3^- + O_2 \qquad \text{(R1)}$$

for which the superoxide anion $O_2^-$ is in equilibrium with its conjugate acid, the hydroperoxyl radical ($HO_2$). This indicates that the in-cloud $O_3$ destruction is sensitive to in-cloud OVOC oxidation. Lelieveld and Crutzen (1990) already proposed that

clouds can influence $HO_x$ ($HO_x=OH+HO_2$), and $NO_x$ ($NO_x=NO+NO_x$), resulting in regional changes of up to 40 % in particular locations, being subject to cloud processing. At the tropics and mid-latitudes, Liang and Jacob (1997) suggest that clouds may reduce $O_3$ by 3 % in summer. By changing the gas-phase oxidant budgets, clouds can indirectly influence the formation of secondary organic aerosol (SOA). Within cloud droplets, OVOC oxidation additionally can lead to the formation and destruction of SOA precursors, and clouds can act as SOA sources (Blando and Turpin, 2000). Further modelling studies suggest that clouds may contribute to the SOA formation on a par with gas-phase sources (Ervens et al., 2011; Lin et al., 2012; Ervens, 2015). By scattering, SOA is known to influence the aerosol optical depth (AOD), leading to a reduction in $NO_2$ photolysis (Tie et al., 2005). In addition, SOA may act as a cloud condensation nuclei (CCN) (Andreae and Rosenfeld, 2008) affecting cloud properties. An increased formation of SOA would thus influence tropospheric $HO_x$ and $O_3$ chemistry.

When performing global modelling studies, it is thus desirable to include the in-cloud oxidation of OVOCs. However, compared to gas-phase chemistry, knowledge of aqueous-phase chemistry still suffers from large uncertainties and most global models only include very limited representations. Most global models include only the uptake of a few soluble compounds, their acid-base equilibria, and the oxidation of sulfur dioxide ($SO_2$) via ozone ($O_3$) and hydrogen peroxide ($H_2O_2$) (Ervens, 2015, Table 1). The explicit oxidation of OVOCs is currently not considered in any global model with one exception though limited to species containing one carbon atom (Tost et al., 2006). Mouchel-Vallon et al. (2017) recently presented the CLoud Explicit Physico-chemical Scheme (CLEPS, version 1.0), a new complex oxidation scheme coupled to the gas-phase Master Chemical Mechanism (MCM, version 3.3.1, Jenkin et al., 2015). However, their comprehensive mechanism is targeted for box-model applications and is not suitable for global model applications, due to its complexity.

In this study, the in-cloud OVOC oxidation scheme Jülich Aqueous-phase Mechanism of Organic Chemistry (JAMOC) is presented and implemented into the chemistry mechanism Module Efficiently Calculating the Chemistry of the Atmosphere (MECCA). Here, JAMOC's representation of organic chemistry is based on CLEPS and is thus an addition to MECCA's existing aqueous-phase chemical mechanism. Therefore, JAMOC needs to be selected by the user upon compilation of MECCA's chemical mechanism. A visualisation of this procedure can be found in MECCA's user manual given in the supplemental material (caaba_mecca_manual.pdf). The modular structure of MECCA allows it to be connected to different base models, e.g., to the Chemistry As A Boxmodel Application (CAABA) by Sander et al. (2019), or to the global ECHAM/MESSy Atmospheric Chemistry Model (EMAC) by Jöckel et al. (2010). In this combination, the proposed mechanism closes the gap between box-models and global model applications. In addition to the new aqueous-phase OVOC chemistry, MECCA also contains the gas-phase Mainz Organic Mechanism (MOM, Sander et al., 2019) with an extensive oxidation scheme for isoprene (Taraborrelli et al., 2009, 2012; Nölscher et al., 2014), monoterpenes (Hens et al., 2014), and aromatics (Cabrera-Perez et al., 2016). VOCs are oxidised by OH, $O_3$ and $NO_3$, whereas $RO_2$ reacts with $HO_2$, $NO_x$, and $NO_3$ and undergoes self- and cross-reactions (Sander et al., 2019).

The mechanism of JAMOC is described in Sect. 2, followed by a short description of its implications in the box-model CAABA (Sect. 3). Global implications are analysed in our companion paper (Rosanka et al., 2020a) and its importance for global models simulating extreme pollution events is addressed by Rosanka et al. (2020b). Modelling uncertainties are discussed in Sect. 4 before drawing final conclusions in Sect. 5.

## 2 The Jülich Aqueous-phase Mechanism of Organic Chemistry (JAMOC)

The detailed mechanism CLEPS includes 850 aqueous-phase reactions, focusing on the oxidation of species containing up to four carbon atoms (Mouchel-Vallon et al., 2017). Since our target is to simulate OVOC chemistry inside the global model EMAC (Rosanka et al., 2020a), using such a large mechanism is not feasible. Therefore, we have developed the reduced mechanism JAMOC. Only a selection of species containing up to four carbon atoms is considered in the aqueous-phase. The gas-phase oxidation of the most abundant hydrocarbons (e.g., methane, isoprene) leads to many highly soluble organic species with one or two carbon atoms (e.g. formaldehyde, methanol, glyoxal). In order to properly represent these degradation products, JAMOC includes the aqueous-phase oxidation of all species containing one and two carbon atoms treated in CLEPS. Even though isoprene ($C_5H_8$), the biogenic VOC emitted the most (Guenther et al., 2012), is not soluble, the representation of its oxidation products containing more than two carbon atoms (i.e. methylgyoxal, methacrolein, methyl vinyl ketone) is desirable for global model applications. Therefore, the oxidation of species containing three carbon atoms in JAMOC focuses on the representation of the aqueous-phase oxidation of methyglyoxal and its aqueous-phase oxidation products (e.g., pyruvic acid). Additionally, its aqueous-phase sources from acetone, hydroxy acetone, isopropanol, hydroperoxide, and isopropyl hydroperoxide are included. The aqueous-phase oxidation of species containing four carbon atoms in JAMOC is limited to methacrolein (MACR) and methyl vinyl ketone (MVK). The phase transfer of species containing up to ten carbon atoms is included so that their wet deposition can be represented in global model applications (i.e. by using EMAC, see Rosanka et al., 2020a). In order to reduce the stiffness of the ODE system and the required computational demand, the representation of organic radicals (see Sect. 2.7) is simplified. It is assumed that the following reactions occur instantly and are not explicitly represented in the ODE system, if it is the only fate of the respective radical: (1) the $O_2$ addition to alkyl radicals, (2) the $HO_2$ elimination of $\alpha$-hydroxyperoxyl, and (3) the carbon bond scission or 1-2 hydrogen shift of alkoxyl radicals. In addition to the chemistry from CLEPS, JAMOC includes: (1) explicit hydration and dehydration, (2) oligomerisation of formaldehyde, glyoxal and methylglyoxal as an in-cloud SOA source, (3) further aqueous-phase photolysis reactions, and (4) the gas-phase photo-oxidation of new outgassed species. The complete aqueous-phase mechanism represents the phase transfer of 368 species, 68 equilibria (acid base and hydration), 402 reactions, and 27 aqueous-phase photolysis reactions. In the gas-phase, 1 photolysis and 18 OH oxidation reactions are added to MOM. A list of the complete mechanism is available in the archived model code.

This section provides a general overview of the developed mechanism. For completeness, short summaries of CLEPS are provided, if no significant difference exists between both mechanisms. Figures 1 and 2 give a graphical representation of all parts of the developed mechanism, using glyoxal and oxalic acid as examples.

### 2.1 Inorganic chemistry

The inorganic chemistry for the proposed mechanism is very similar to the inorganic chemistry of the standard aqueous-phase mechanism used in EMAC (Tost et al., 2007; Jöckel et al., 2016). In this standard mechanism, the major aqueous-phase $O_3$ sink, the reaction with $O_2^-$, is represented as:

$$O_3 + O_2^- \rightarrow OH + OH^- \tag{R2}$$

In JAMOC, this aqueous-phase $O_3$ chemistry is updated to the mechanism proposed by Staehelin et al. (1984) with corrections from Staehelin and Hoigné (1985), in which the $O_3$ destruction by $O_2^-$ is represented as given in Reaction R1.

## 2.2 Uptake of gaseous species into cloud droplets

The mass transfer of species between the gas- and the aqueous-phase is described following Schwartz (1986) (see Sander, 1999; Tost et al., 2006). The explicit bidirectional phase transfer of 45 carbon-containing species, which explicitly react in the aqueous-phase, is considered (indicated in pink in Fig. 1 and 2). In this model framework, Henry's law constants are mainly taken from Sander (2015), Burkholder et al. (2015), and sources therein. In order to account for the hydration of aldehydes (for more details see Sect. 2.3), a distinction is made between the effective Henry's law constant ($H^*$) and the intrinsic Henry's law constant ($H$). The latter is calculated by:

$$H = H^*/(1 + K_{hyd}) \tag{1}$$

where $K_{hyd}$ is the ratio between the forward and reverse kinetic rate constant of the hydration equilibrium (see Reaction R3). Table 1 gives an overview of the hydration constants and the effective Henry's law constants, including the resulting intrinsic Henry's law constants, for all aldehydes. The temperature dependencies of the intrinsic Henry's law constants are assumed to be the same as for the effective constants. The accommodation constant ($\alpha$) is known for a few species, if unknown the standard EMAC estimate of 0.1 is used. In addition to the phase transfer of all species that explicitly react in the aqueous-phase, the phase transfer of all soluble MOM species containing up to ten carbon atoms is represented in order to allow their removal by wet deposition in global models (i.e. by using EMAC, see Rosanka et al., 2020a). A list summarising all Henry's law and accommodation constants is available in the archived model code.

## 2.3 Hydration of carbonyls

Gem-diols are formed when aldehydes (carbonyl compounds) hydrate:

$$R_2C{=}O + H_2O \rightleftharpoons R_2C(OH)(OH) \tag{R3}$$

In the new mechanism, 12 carbonyl species undergo hydration (indicated with blue arrows in Fig. 1). The monohydrate of glyoxal (dihydroxyacetaldehyde) undergoes an additional hydration to form its dihydrate (1,1,2,2–Ethanetetrol). Pseudo-first order rate constants for the hydration and dehydration are mainly obtained from the literature (e.g. Doussin and Monod, 2013). In the case of formyldioxidanyl and hydroperoxyacetaldehyde the pseudo-first order rate constants are assumed to be the same as for formaldehyde and glycolaldehyde, respectively.

The typical lifetime of a warm cloud droplet can be several minutes but their typical evaporation timescale is less than 100 s (Jarecka et al., 2013). Following the dehydration constants presented by Doussin and Monod (2013), the dehydration of some gem-diols can be slower than the typical cloud droplet evaporation timescale. Additionally, their rapid transfer across the phases is expected to affect the gas-phase concentration of gem-diols, for which no other significant source is known. This process could be an important removal of gem-diols from the aqueous-phase, without yielding the original aldehyde. Therefore, their

outgassing is considered for use with the models representing evaporating clouds like the EMAC model (following Sect. 2.2). However, their Henry's law constants are unknown. Thus, estimates are obtained at 25 °C using the bond method (Meylan and Howard, 1991) from the United States Environmental Protection Agency Estimation Programs Interface (EPI) Suite (US EPA,

2012). An overview of all estimated effective Henry's law constants is given in Table 2.

In CLEPS, acyl peroxy radicals ($RC(O)(OO)$) are assumed to be in a hydration/dehydration equilibrium similar to their parent aldehydes (Mouchel-Vallon et al., 2017). However, experimental results by Villalta et al. (1996) show that in the case of peroxyacetyl radicals ($CH_3C(O)(OO)$) no equilibrium exists. Instead, hydrolysis takes place likely yielding acetic acid ($CH_3CO_2H$) and $HO_2$. It is thus assumed that all acyl peroxy radicals undergo hydrolysis following:

(R4)

with a reaction rate constant of $7.0 \times 10^5 \, \mathrm{M^{-1}s^{-1}}$, as proposed by Villalta et al. (1996).

## 2.4  Acid dissociation

The dissociation of acids is taken into account following:

$$R_2CO(OH) \rightleftharpoons R_2CO(O^-) + H^+$$ (R5)

which is indicated in green in Fig. 2. The acidity constants ($K_a$) for most one, two, and three carbon containing acids taken into account in JAMOC, are known from the literature (Rumble, 2020). If unknown, the acidity constants are used as proposed by Mouchel-Vallon et al. (2017). The dissociation and association rate constants are selected such that the equilibrium between dissociation and association is reached quickly, while still avoiding numerical stiffness problems in the numerical integrator.

## 2.5  Oxidation by $OH$, $NO_3$ and other oxidants

In JAMOC, $OH$ and $NO_3$ are the main oxidants taken into account. Reactions of OVOCs with oxidants are treated as proposed by Mouchel-Vallon et al. (2017). Organic compounds may react in three different ways with OH radicals (Herrmann et al., 2015) each indicated in orange in Fig. 1 and 2. They form an alkyl radical following H-abstraction:

$$RH + OH \rightarrow R + H_2O$$ (R6)

If the organic compound contains a double bond OH-addition is favoured:

(R7)

With anions like carboxylates electron transfer takes place:

(R8)

When available, rate constants are obtained from literature. If unavailable, the rate constant for the H-abstraction are estimated based on the structure activity relationship (SAR) by Doussin and Monod (2013), which for carboxylate compounds is extended to account for the electron transfer as described by Mouchel-Vallon et al. (2017). In all cases, branching ratios are obtained from the SAR with simplifications by Mouchel-Vallon et al. (2017).

During nighttime, OH radical concentrations are low and due to missing photolysis, $NO_3$ radicals are considered to be the main nighttime oxidant. Similar to CLEPS, JAMOC only considers the H-abstraction leading to alkyl radicals for $NO_3$ reactions (Herrmann et al., 2015):

$$RH + NO_3 \rightarrow R + NO_3^- + H^+ \tag{R9}$$

For most species containing one or two carbon atoms, rate constants are obtained from the literature. Opposite to OH, no SAR is available for the H-abstraction by $NO_3$. Therefore, rate constants are obtained from the similar criteria described by Mouchel-Vallon et al. (2017). Due to missing branching ratios from the literature, branching ratios are assumed to be the same as for the H-abstraction by OH.

In addition to reactions of organic compounds with OH and $NO_3$, reactions with other oxidants are implemented when available from the literature. The oxidants considered here are $O_2^-$, $O_3$, $H_2O_2$, $CO_3^-$, and sulfur containing oxidants ($SO_4^-$ and $SO_5^-$). For all oxidation reactions, reaction rates and branching ratios are either taken from literature or as proposed by Mouchel-Vallon et al. (2017).

### 2.6 Oligomerization

The formation of oligomers within the atmospheric aqueous phase is known to be a source of SOA. Even though Tan et al. (2009) suggest that the formation of oligomers becomes increasingly important for aerosol water, where precursor concentrations are found to be higher, Lin et al. (2012) demonstrated that SOA formation from cloud processing is globally important. Therefore, JAMOC includes self- and cross-reactions leading to oligomers for formaldehyde, glyoxal, and methylglyoxal. The oligomerization of formaldehyde is implemented following Hahnenstein et al. (1995), in which the methanediol formed from hydrolosis (see Sect. 2.3) reacts with itself and the dimer formed from this self-reaction. Ervens and Volkamer (2010) studied the oligomerization of glyoxal. Here, glyoxal and its hydrates react with the monohydrate to form three oligomers (indicated in green in Fig. 1). The oligomerization of methylglyoxal is assumed to follow the same mechanisms as for glyoxal. However, only the monohydrate of methylglyoxal is taken into account in this mechanism, leading to only two oligomers. Each oligomer is assumed to react with OH, leading to $HO_2$, with reaction rate constants that are double for the corresponding (hydrated) monomer due to increased number of abstractable H-atoms.

### 2.7 Organic radicals

Organic radicals are generally treated following Mouchel-Vallon et al. (2017). Alkyl radicals can either form oligomers via self- and cross-reactions (e.g. Lim et al., 2013; Ervens et al., 2015) or undergo $O_2$ addition:

$$R + O_2 \rightarrow R(OO) \tag{R10}$$

As proposed by Mouchel-Vallon et al. (2017), it is assumed that $O_2$ addition is the fastest pathway, due to high $O_2$ concentrations following a fast $O_2$ saturation in cloud droplets (Ervens, 2015). Thus, oligomers formed from the self- and cross-reactions of alkyl radicals are not considered in JAMOC.

    Peroxyl radicals generally undergo self- or cross-reactions forming short-lived tetroxides that quickly decompose (von Sonntag and Schuchmann, 1997). Due to limited computation resources, only self-reactions are taken into account. Mouchel-
Vallon et al. (2017) propose three similarity criteria for the decomposition of tetroxides depending on the peroxyl radical: (1) for $\beta$-peroxycarboxylic acids ($RC(OO)C(=O)(OH)$) experimental results from Schuchmann et al. (1985) are generalised, (2) $\beta$-hydroxyperoxyl radicals ($>C(OH)C(OO)<$) are represented according to Piesiak et al. (1984), and (3) $\beta$-oxoperoxyl radicals ($-COC(OO)<$) are treated based on Zegota et al. (1986) and Poulain et al. (2010). If some products are unknown, branching ratios of the known products are rescaled to 100 % in order to preserve mass. The peroxyl radicals undergo $HO_2$
elimination (von Sonntag, 1987), if the hydroxyl moiety is in the alpha position ($\alpha$-hydroxyperoxyl):

(R11)

The generalised corresponding rate constants are used as proposed by Mouchel-Vallon et al. (2017, Table 3), which are based on the work of von Sonntag (1987). In CLEPS, peroxyl radicals additionally undergo $O_2^-$ elimination when reacting with $OH^-$ (Zegota et al., 1986; Mouchel-Vallon et al., 2017):

(R12)

In order to decrease the number of reactions and due to the fast $HO_2$ elimination, this $O_2^-$ elimination is not considered explicitly in JAMOC.

    Acyl peroxy radicals ($RC(O)(OO)$) are treated like peroxyl radicals, as described in Monod et al. (2007), but only form alkoxyl radicals. Peroxyl radicals that have not explicitly discussed so far are treated following Monod et al. (2007) (Mouchel-
Vallon et al., 2017).

    Mouchel-Vallon et al. (2017) suggest that alkoxyl radicals (RO) either undergo a carbon bond scission (Hilborn and Pincock, 1991), if the neighbouring carbon atom is oxygenated:

(R13)

or an 1-2 hydrogen shift (DeCosta and Pincock, 1989), if the neighbouring carbon atom is not oxygenated:

(R14)

## 2.8 Photolysis

In general, the photolysis of some organic compounds (e.g. organic peroxides, pyruvic acid) competes with other oxidation pathways (see Sect. 2.5) and can be a source of OH. In Rosanka et al. (2020a), a global tropospheric in-cloud OH budget is presented. When using JAMOC, EMAC predicts that about 40 % of all in-cloud OH is produced from the photolysis of a selection of organic compounds. However, Fenton chemistry is not considered by Rosanka et al. (2020a) and the relative contribution is therefore expected to be overestimated. The photolysis of glyoxal and oxalic acid is indicated in orange in Fig. 1 and 2. The number of photolytic reactions known from literature, of which some are implemented in CLEPS (Mouchel-Vallon et al., 2017), is limited. In JAMOC, the photolysis of additional compounds is taken into account. This includes the photolysis of oxalic acid ($(COOH)_2$), which is implemented following Yamamoto and Back (1985) using the ultraviolet absorption spectrum presented in Back (1984). If available, additional photolysis reactions are implemented following Sander et al. (2014). In order to account for scattering effects within cloud droplets (Ruggaber et al., 1997), an enhancement factor of 2.33, the same as used in EMACs standard aqueous-phase mechanism for the photolysis of $H_2O_2$ (Tost et al., 2007; Jöckel et al., 2016), is applied to each gas-phase photolysis rate.

## 2.9 Gas-phase oxidation of new species

Oxalic acid was not represented in the gas-phase mechanism (i.e. in MOM). The gas-phase oxidation of oxalic acid via OH and its photolysis are implemented, in order to realistically represent oxalic acid in the gas-phase. Similar to the implementation in the aqueous-phase, the photolysis of oxalic acid is implemented following Yamamoto and Back (1985) and Back (1984). All gem-diols (see Sect. 2.3) formed from hydration are transferred to the gas-phase and oxidised via OH (indicated in orange in Fig. 1 and 2). All OH oxidation reaction rates are estimated following the description of Sander et al. (2019).

## 3 Influence of JAMOC on a single air parcel

The implications of the developed mechanism are tested by comparing it to the minimum in-cloud oxidation scheme available in CAABA/MECCA and EMAC. The minimum mechanism only includes the uptake of a few soluble compounds, their acid-base equilibria, and the oxidation of $SO_2$ via $O_3$ and $H_2O_2$ (Jöckel et al., 2006). This minimal mechanism is thus representable for most global models (Ervens, 2015). For both mechanisms, an air parcel is simulated in CAABA taking the same conditions into account: the air parcel is simulated during summer at mid-latitude with a constant temperature of 278 K and relative humidity of 100 %. Table 3 provides a selection of initial mixing ratios and emission fluxes of gas-phase species treated in MOM. The initial conditions are a modified version of the scenario used by Taraborrelli et al. (2009). Within the air parcel, a stable cloud droplet population is simulated with a radius of 20 μm and a liquid water content of 0.3 g m$^{-3}$. Both simulations are intended as a sensitivity study of JAMOC. Therefore, CAABA is initialised at 0 UTC and simulates the air parcel for five days in total. A realistic cloud event with a cloud droplet lifetime of one hour using CAABA is presented in Rosanka et al. (2020a). In addition, Rosanka et al. (2020a) study the implications of JAMOC on a global scale using EMAC.

Figure 3 gives an overview of the temporal development of the total mixing ratios (gas- + aqueous-phase) for a selection of species during the simulated daily cycles of five days. Comparing the new and the minimum mechanisms, it becomes clear that the new developed mechanism has a significant impact on most trace gases. With the explicit oxidation of many OVOCs in the
240 aqueous-phase, the mixing ratio of the sum of all OVOCs explicitly reacting in JAMOC ($\sum$OVOCs, see Equation A1 in Appendix A) are significantly reduced. This reduction is a combined effect from: (1) the in-cloud oxidation of these OVOCs, and (2) their dampened gas-phase production. In the gas-phase, most OVOCs are formed by secondary production (e.g. oxidation of primarily emitted VOCs). The decrease of the main VOC oxidant (i.e. OH) leads to a reduced oxidation of primarily emitted VOCs resulting in a reduced gas-phase OVOC formation. The calculated diurnal cycles of OH, $HO_2$, $NO_x$, and $O_3$ are similar
for both mechanisms and differ mainly in the absolute mixing ratios calculated. When JAMOC is used, $HO_2$ partitions into the cloud droplets, whereas NO stays in the gas-phase due to its low solubility (Jacob, 1986; Lelieveld and Crutzen, 1990). This results in substantial changes in the $NO_x$–$HO_x$ relation, resulting in a reduced OH formation from its second most important atmospheric gas-phase source:

$$NO + HO_2 \rightarrow NO_2 + OH \tag{R15}$$

Overall, this results in reduced $HO_x$ and elevated $NO_x$ mixing ratios, respectively. In addition, lower $HO_2$ mixing ratios lead to a reduced removal of $NO_x$ by the formation of nitric acid ($HNO_3$) and peroxynitric acid ($HNO_4$). Within the cloud droplet, $O_2^-$ is in equilibrium with its conjugated base $HO_2$. Higher in-cloud $HO_2$ concentrations, caused by mass transfer and in-cloud OVOC oxidation, consequently lead to an increased destruction of $O_3$ via Reaction R1. This results in an enhanced uptake of $O_3$ into the cloud droplet and an increased importance of cloud droplets as $O_3$ sinks.

The impact of the new proposed mechanism is consistent with earlier box-model studies. The reduction in OVOCs is similar to the findings given in Mouchel-Vallon et al. (2017) when using CLEPS. In contrast, the reduction in methylglyoxal differs since in CLEPS, gas-phase methylglyoxal mixing ratios first increase and later decrease during the modelled cloud event of Mouchel-Vallon et al. (2017). This difference is most likely linked to the usage of the intrinsic Henry's law constant and the explicit representation of the methylglyoxal hydration/dehydration in JAMOC. Opposite to Mouchel-Vallon et al. (2017),
CAABA predicts a reduction in OH levels. However, this reduction in OH is in line with other modelling studies predicting a similar reduction of gas-phase OH during cloud events (Tilgner et al., 2013). It is important to keep in mind that in Mouchel-Vallon et al. (2017), a different cloud event is simulated, including different initial conditions and a different emission scenario. In their study, the cloud forms after a certain time period, whereas in CAABA the cloud is present the whole time.

## 4  Model uncertainties

The uncertainties associated with the present kinetic model are mainly attributed to (1) assumptions and simplifications in the aqueous-phase mechanism, and (2) missing sinks of key oxidants. Each possible uncertainty is discussed in this section.

In general, aqueous-phase kinetics data suffer from many large uncertainties compared to the data available for the gas-phase. In the development of the implemented in-cloud oxidation scheme JAMOC, some assumptions are made that introduce

modelling uncertainties. If rate constants are unknown, estimates are taken from Mouchel-Vallon et al. (2017). These are based on a structure-activity relationship (SAR) for the H-abstraction by OH for dissolved carbonyls and carboxylic acids considered in this study (Doussin and Monod, 2013). However, it is expected that the uncertainty in the estimated rate constants is low since Doussin and Monod (2013) report that when evaluated using experimental data, their estimates were within $\pm 20\,\%$ for 58 % of the calculated rate constants. Also the up-scaling of branching ratios to conserve mass, further influences the predictions of VOC oxidation. The mechanism should be updated with rate constants and branching ratios as soon as experimental results become available. The increased concentration and burden of certain organic acids heavily depend on the chemistry and solubility of some gem-diols. For example, the gas-phase oxidation of the methylglyoxal monohydrate leads to the formation of pyruvic acid. The gas-phase production of pyruvic acid therefore depends on the mass transfer of this specific monohydrate. In the current implementation, the Henry's law constants for all gem-diols are estimated. For the methylglyoxal monohydrate, the estimated values range from $3.5 \times 10^3\,\mathrm{M\,atm^{-1}}$ to $2.4 \times 10^4\,\mathrm{M\,atm^{-1}}$.

Phase-transfer of soluble VOCs into cloud droplets is considered in JAMOC even when their oxidation is not explicitly represented (see Sect. 2.2). This allows their removal from the atmosphere by rain-out when JAMOC is connected to a global model (e.g. using EMAC, see Rosanka et al., 2020a). Arakaki et al. (2013) point out that by not taking the oxidation of all dissolved organic carbon (DOC) into account, aqueous-phase OH concentrations might be overestimated. Based on observational estimates, they suggest a general scavenging rate constant of $k_{C,OH} = (3.8 \pm 1.9) \times 10^8\,\mathrm{M^{-1}\,s^{-1}}$ for all DOC. If each DOC reacts with OH, the gas-phase concentration would be reduced, further influencing gas-phase VOC concentrations and the overall oxidation capacity. Implementing the DOC oxidation, suggested by Arakaki et al. (2013), for every scavenged DOC species would increase the aqueous-phase mechanism by more than 280 reactions, which is almost a doubling of the proposed organic mechanism. Within the scope of this study, it is thus computationally not feasible to include this additional OH sink. Currently, the model run time increases from 4.3 s for EMACs minimum in-cloud oxidation scheme to 6.5 s for the new proposed mechanism JAMOC.

Reducing the model uncertainties introduced by estimates of Henry's law constants of gem-diols, and missing in-cloud DOC oxidation, is outside the scope of this study due to their complexity. Model representation of the latter is expected to influence the oxidation rate of VOCs in the cloud droplets and aerosols.

## 5 Conclusions

In this study, the new in-cloud oxidation scheme of soluble VOCs JAMOC is developed and implemented into MECCA. This mechanism is suitable for global model applications and based on the box-model mechanism CLEPS proposed by Mouchel-Vallon et al. (2017). The mechanism considers the phase transfer of OVOCs containing up to ten carbon atoms. For a selection of OVOCs containing up to four carbon atoms, their acid/base and/or hydration/dehydration equilibria, and their reactions with OH, $NO_3$ and other oxidants (if available) are explicitly represented. Additionally, the gas-phase photo-oxidation of gem-diols and oxalic acid was implemented into the gas-phase mechanism MOM. Finally, JAMOC was tested within the CAABA box-model.

The proposed mechanism leads to a significant reduction in OVOCs and an overall reduction in important oxidants. These findings are in line with other box-model studies and demonstrate the importance of in-cloud chemistry in atmospheric chemistry. By not taking the in-cloud oxidation of OVOCs into account, global models will tend to overestimate the levels of OVOCs and atmospheric oxidants. A complete analysis on the importance of JAMOC at a global scale is presented in Rosanka et al. (2020a). In future studies, the modular implementation of JAMOC, with the necessary adjustments, allows its application to aerosol water.

*Code and data availability.* The current version of the CAABA/MECCA model code is available as a community model in the code repository at https://gitlab.com/RolfSander/caaba-mecca (last access: 23 April 2021), published under the GNU General Public License (http://www.gnu.org/copyleft/gpl.html, last access: 23 April 2021). The exact version of the CAABA/MECCA model (version 4.5.0) developed in this paper and used in each simulation presented in this paper is archived at Zenodo (http://doi.org/10.5281/zenodo.4707938, Sander, 2021). All future versions of CAABA/MECCA will be made available at https://doi.org/10.5281/zenodo.4707937. The archived model code includes a list of all chemical reactions including rate constants and references (caaba/manual/meccanism.pdf), a list of all Henry's law and accommodation constants (caaba/tools/chemprop/chemprop.pdf), and a user manual (caaba/manual/caaba_manual_manual.pdf). The model output of all simulations presented in this paper are archived at Jülich DATA (https://doi.org/10.26165/JUELICH-DATA/SD9F6B, Rosanka et al., 2021). For further information and updates, the CAABA/MECCA web page at http://www.mecca.messy-interface.org (last access: 23 April 2021) can be consulted.

## Appendix A: Definition of $\sum$OVOCs

In Fig. 3, the mixing ratios of the sum of all the OVOCs explicitly reacting in JAMOC ($\sum$OVOCs) are shown. In these cases, $\sum$ OVOCs is defined as:

$$\sum\text{OVOCs} = \text{methanol} + \text{formaldehyde} + \text{methyl hydroperoxide} + \text{hydroxymethylhydroperoxide} + \text{ethanol} +$$
$$\text{ethylene glycol} + \text{acetaldehyde} + \text{glycolaldehyde} + \text{glyoxal} + \text{1-hydroperoxyacetone} +$$
$$\text{methylglyoxal} + \text{isopropanol} + \text{isopropyl hydro peroxide} + \text{methacrolein} + \text{methyl vinyl ketone}$$

$$(A1)$$

*Author contributions.* SR and DT developed the chemical mechanism. The chemical mechanism was reviewed by RS. SR, DT, and RS implemented the mechanism into MECCA. The results were discussed by all co-authors. The manuscript was prepared by SR with the help of all co-authors.

*Competing interests.* The authors declare that they have no competing of interest.

*Acknowledgements.* The work described in this paper has received funding from the Initiative and Networking Fund of the Helmholtz Association through the project "Advanced Earth System Modelling Capacity (ESM)". The content of this paper is the sole responsibility of the author(s) and it does not represent the opinion of the Helmholtz Association, and the Helmholtz Association is not responsible for any use that might be made of the information contained. The authors gratefully acknowledge the Earth System Modelling Project (ESM) for funding this work by providing computing time on the ESM partition of the supercomputer JUWELS at the Jülich Supercomputing Centre (JSC).

**Table 1.** Hydration constants ($K_{\mathrm{hyd}}$), effective ($H^*$) and intrinsic ($H$) Henry's law constants for aldehydes (see Sect. 2.2 for details). If not stated otherwise, hydration constants are obtained from Doussin and Monod (2013) and sources therein. If not stated otherwise, effective Henry's law constants are taken from Burkholder et al. (2015).

| Species | $K_{\mathrm{hyd}}$ | $H^*$ [M/atm] | $H$ [M/atm] |
|---|---|---|---|
| Formaldehyde | 1278.0 | $3.23 \times 10^3$ | 2.53 |
| Acetaldehyde | 1.2 | $1.29 \times 10^1$ | 5.91 |
| Glycolaldehyde | 15.7 | $4.00 \times 10^4$ | $2.40 \times 10^3$ |
| Glyoxal | 350.0 [a] | $4.19 \times 10^5$ | $1.19 \times 10^3$ |
| Glyoxylic acid | 1100.0 | $1.09 \times 10^4$ | 9.90 |
| Methylglyoxal | 2000.0 | $3.50 \times 10^{3,\,b}$ | 1.75 |

[a] Ervens and Volkamer (2010)

[b] Betterton and Hoffmann (1988)

**Table 2.** Estimated effective ($H^*$) Henry's law constants for all gem-diols represented in JAMOC. Estimates with the bond method (Meylan and Howard, 1991) are obtained from US EPA, 2012.

| Species | $H^*$ [M/atm] |
|---|---|
| Methanediol | $1.02 \times 10^4$ |
| 1,1–Ethanediol | $7.63 \times 10^3$ |
| Dihydroxyacetaldehyde | $2.58 \times 10^3$ |
| 1,1,2,2–Ethanetetrol | $5.71 \times 10^6$ |
| 2,2-Dihydroxyacetic acid | $3.21 \times 10^5$ |
| 1,1,2–Ethanetriol | $2.09 \times 10^5$ |
| Hydroperoxyacetaldehyde hydrate | $2.09 \times 10^5$ |
| 1,1-Dihydroxyacetone | $3.53 \times 10^3$ |

**Table 3.** Initial box-model (CAABA) mixing ratios and emission rates for selected gas-phase species. Initial mixing ratios are a modified version of the scenario used by Taraborrelli et al. (2012).

| Gas-phase species | Initial mixing ratio [nmol/mol] | Emission [mole. $cm^{-2}$ $s^{-1}$] |
|---|---|---|
| $O_3$ | 30 | - |
| NO | 0.01 | $3.3 \times 10^{-9}$ |
| $NO_2$ | 0.1 | - |
| $HNO_3$ | $5.0 \times 10^{-3}$ | - |
| $H_2O_2$ | 7 | - |
| CO | 100 | - |
| $CO_2$ | $3.5 \times 10^{5}$ | - |
| $CH_4$ | $1.8 \times 10^{3}$ | - |
| Formaldehyde | 5 | - |
| Methanol | 0.5 | - |
| Methyl peroxide | 4 | - |
| Formic acid | 0.35 | - |
| Acetic acid | 2 | - |
| Peroxy acetic acid | 1.5 | - |
| Hydroxy acetone | 4 | - |
| Methylglyoxal | 0.5 | - |
| Isoprene | 0.1 | - |
| Peroxyacetylnitrate | 0.1 | - |
| Ethane | 2 | - |

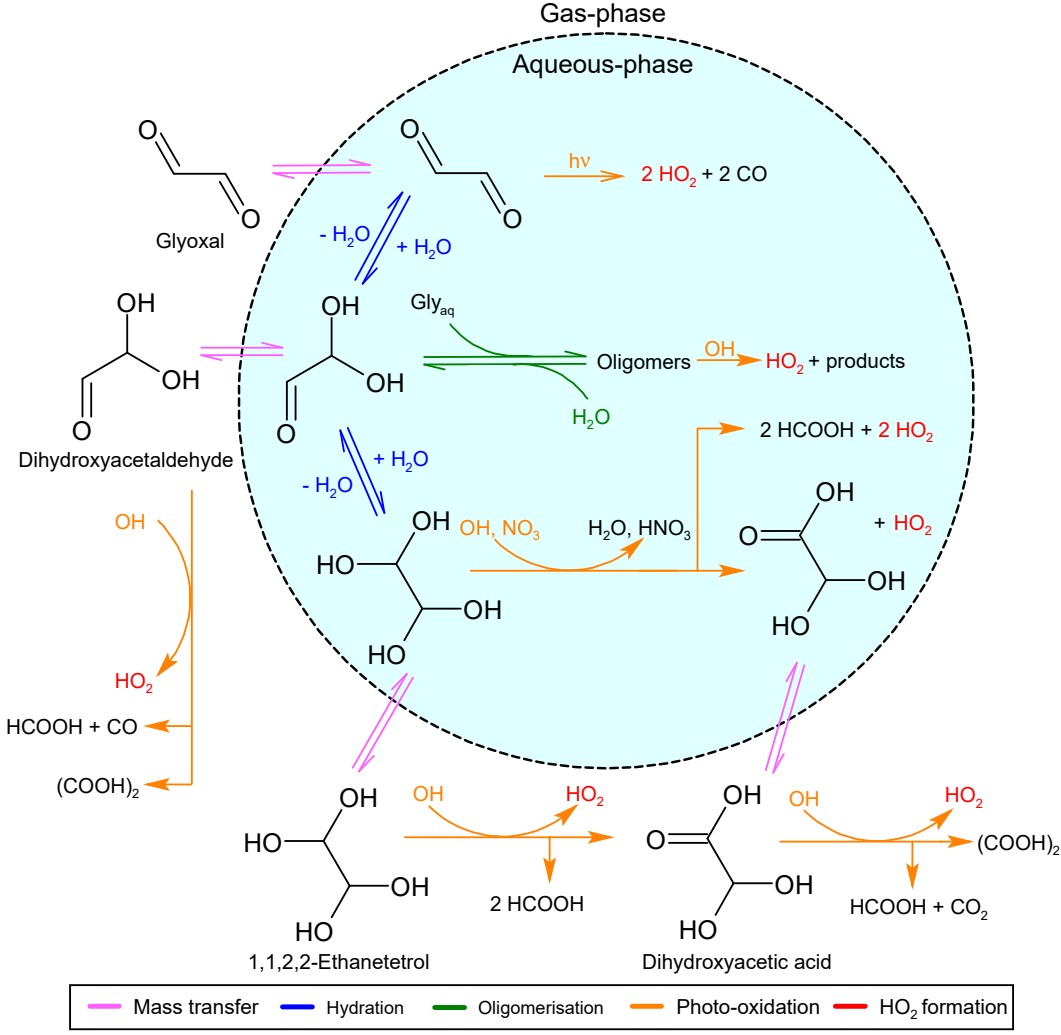

**Figure 1.** Oxidation of glyoxal ($CHOCHO$) by radicals in JAMOC. The oligomerisation of the glyoxal monohydrate occurs with glyoxal as well as with its hydrates (see Sect. 2.6). Here, $Gly_{aq}$ denotes all three forms of glyoxal (glyoxal, its monohydrate, and its dihydrate), which is consistent with the kinetic data published by Ervens and Volkamer (2010). $(COOH)_2$ denotes oxalic acid whose representation in JAMOC is illustrated in Fig. 2. The following aspects are not explicitly represented: (1) the oxidation of the glyoxal dihydrate via the sulfate radical anion ($SO_4^-$), (2) the aqueous-phase sources of glyoxal and the glyoxal monohydrate from the oxidation of glycolaldehyde and the glycolaldehyde monohydrate, and (3) the aqueous-phase oxidation of dihydroxyacetic acid.

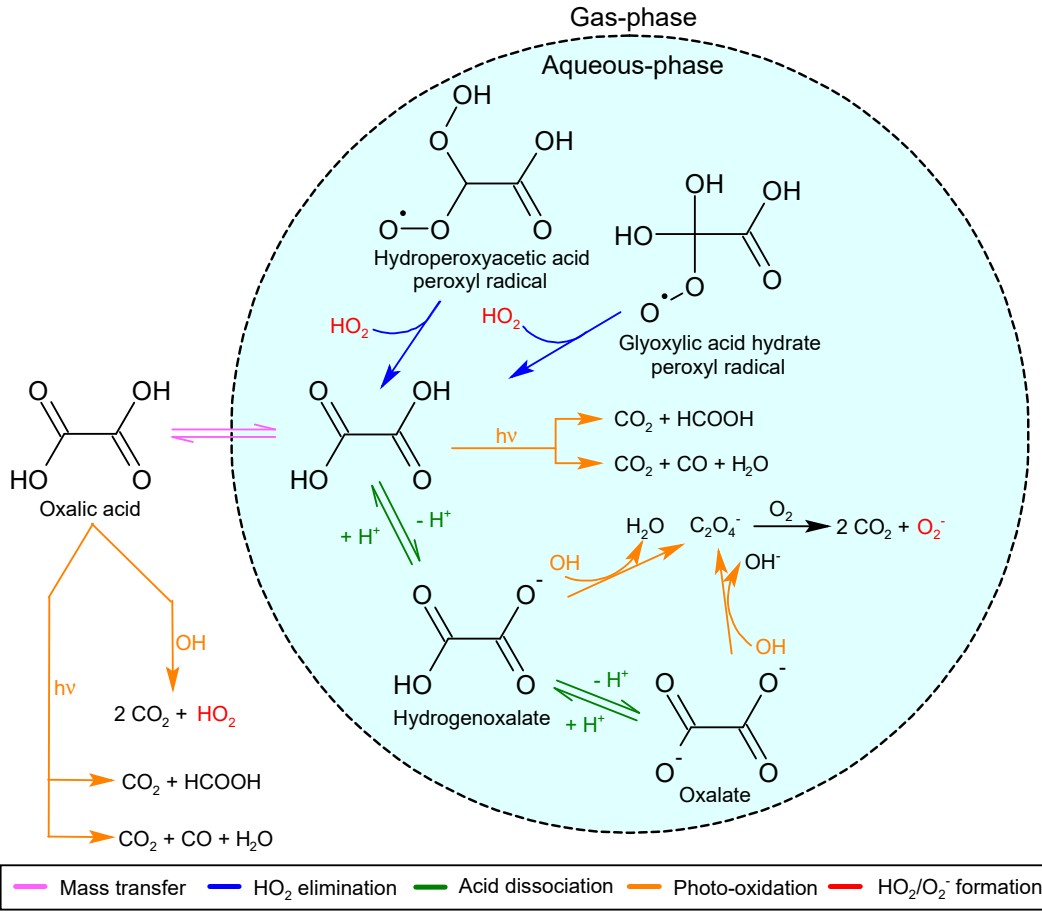

**Figure 2.** Formation and oxidation of oxalic acid ($(COOH)_2$) by radicals in JAMOC. The oxidation via the sulfate radical anion ($SO_4^-$) is not shown.

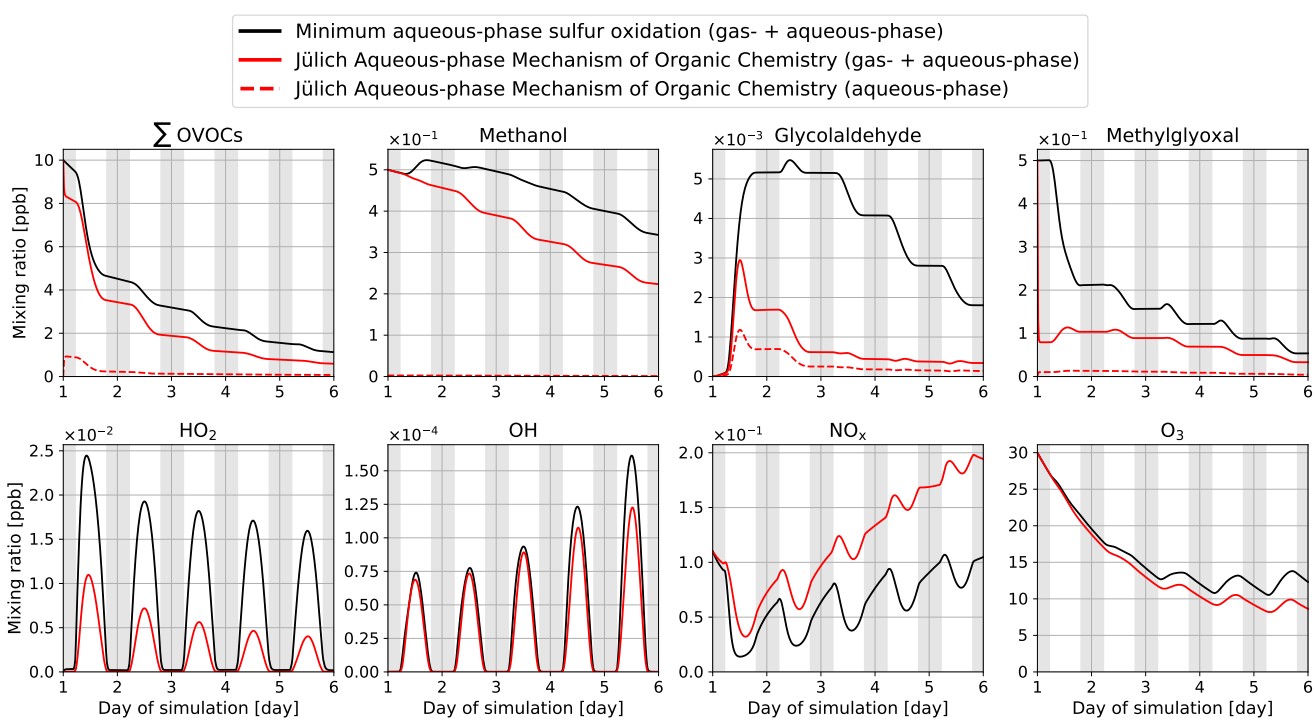

**Figure 3.** Time evolution for total mixing ratios (gas- + aqueous-phase) of the sum of all the OVOCs explicitly oxidised in the proposed mechanism ($\sum$OVOCs, see Equation A1 in Appendix A), methanol, glycolaldehyde, methylglyoxal, $HO_2$, OH, $NO_x$, and $O_3$ within the boxmodel CAABA. Mixing ratios are provided for two cases, one using the minimum aqueous-phase mechanism in global models (sulfur oxidation only, black line) and JAMOC (red line). In addition, aqueous-phase mixing ratios of $\sum$OVOCs, methanol, glycolaldehyde, and methylglyoxal are given for the simulation using JAMOC. The aqueous-phase mixing ratios include the gem-diols formed for the species listed in Table 1. Nighttime is indicated by a grey background shading. Note that lines may overlap.

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
