# Peer review of "Oxidation of low-molecular weight organic compounds in cloud droplets: development of the JAMOC chemical mechanism in CAABA/MECCA (version 4.5.0gmdd)"

_Geoscientific Model Development, 2020_

## Referee Comment (RC1) · Anonymous Referee #1 · 20 Nov 2020

Review of "Oxidation of low-molecular weight organic compounds in cloud
droplets: development of the JAMOC chemical mechanism in CAABA/MECCA (version 4.5.0gmdd)
Author(s): Simon Rosanka et al.

The authors present a new chemical mechanism (JAMOC), implemented in MECCA. JAMOC includes explicit oxidation steps of organic compounds in the aqueous phase of cloud droplets and thus exceeds previous aqueous phase chemistry mechanisms suitable for box, regional and global modeling. Such extensions are urgently needed as currently, particularly in global models, aqueous phase chemistry modules are largely limited to sulfur(IV) oxidation.
However, the current manuscript needs major clarifications and additions to make it a comprehensive and useful extension to previous multiphase model studies. The choice of the newly added reactions is not always evident and also the discussion of the example results in Figure 3 is misleading. In addition, at several places, terminology is confusing or inaccurate. Also previous literature on atmospheric multiphase modeling should be properly discussed. Overall, while I think that this reaction mechanism could possibly become a useful addition to the currently used ones, my comments below need to be carefully addressed prior to possible recommendation for publication.

**Major comments**

1. Oligomerization has been discussed to be only relevant in the aqueous phase of aerosol particles where organic concentrations may be higher than in cloud droplets (Ervens et al., 2015; Perri et al., 2009; Tan et al., 2009). Thus, it is not clear why they are included in the mechanism. While aqueous phase reactions might also occur in the aqueous phase of aerosol particles, such an extension would also need to include adjustments of Henry's law constants and kinetic reaction rate constants for high ionic strength. Thus, this seems out of the scope of the current study.

2. The authors discuss at several places the role of gem-diols and the need of including their phase partitioning between gas and aqueous phases. Is there any indication of the relevance of such processes? Given the very small concentration of water in the gas phase, the stability of gem-diols in the gas phase is likely very small. I assume that their gas phase fraction is likely negligible. Unless the authors can provide literature or estimates on their Henry's law constants that show the opposite, I would not identify the inclusion of their partitioning into atmospheric chemical mechanisms as one of the main gaps in current mechanism developments. In fact, the hydrated glyoxylic acid has been shown to be of such low volatility that it can be involved in new particle formation (Liu et al., 2017)
    - Please show the estimated Henry's law constants in Table 1.
    - Add references that indicate their potential relevance in the gas phase.

3. At several places in the manuscript, it is not clear whether the authors refer to predictions of aqueous phase rates and budgets or to both phases. For example,
    - l. 59: R1 is certainly not a major sink in the atmosphere, but only in the aqueous phase
    - l. 190: is the explicit oxidation of OVOCs only added in the aqueous or also in the gas phase?

4. At several places in the manuscript, misleading or wrong terminology is used, e.g.
    - l. 38: 'nitrogen trioxide' is usually referred to as 'nitrate radical'
    - l. 50: 'simulating hydration and dehydration explicitly' implies that the hydration reaction and dehydration reactions are implemented. However, it seems that only the equilibria are included, separately from the gas/aqueous phase partitioning described by Henry's law.

- l. 134: Do you mean 'recombination of alkyl radicals', i.e. the self-reaction of two radicals?

5. The model studies are performed for a period of 5 days. However, the typical lifetime of a cloud droplet is on the order of 30 min or less (i.e. the time a droplet spends between cloud base and top in up- and downdraft regimes). You should at least mention that the model simulations are highly idealized and should be regarded as a sensitivity study rather than a realistic scenario.

6. The authors imply that the additional OVOC oxidation in the aqueous leads to the significant decrease in predicted OH concentration. However, the main reason why significantly less OH is observed in the presence of clouds is the decrease in its formation rate in the gas phase from the reaction of $NO + HO_2$. These reactants are separated due to their significantly different water-solubility, e.g. (Jacob, 1986; Lelieveld and Crutzen, 1990). Consequently, the lower $HO_2$ and higher NO levels in the presence of clouds are not due to the different water solubilities of OH and $HO_2$ but because of the differences in the gas phase photochemical cycles of $HO_x$ and $NO_x$. This should be discussed and analyzed in Section 3.

7. Given that currently only a small fraction (~ 15%) of organics in cloud water can be identified on a molecular basis, (e.g., Herckes et al., 2013), implies that also even the most detailed chemical aqueous phase mechanism is likely largely incomplete in terms of organic species. Thus, also the predicted OH(aq) concentration is likely biased high as by far not all sinks are included. The idea of the general scavenging rate constant as suggested by Arakaki et (2008) is that it can applied to parameterize the loss of OH(aq). Thus, it would seem a reasonable 'short-cut' to implement water-soluble organic carbon (WSOC) mass as an additional 'species' in the mechanism that reacts with OH(aq) to account for missing OH(aq) sources. As the products may be in many cases other WSOC compounds, this reaction could be implemented as WSOC + OH → WSOC + $HO_2$ (k = 3.8e8 M-1 s-1). How would the implementation of this reaction change the results in general, and in particular the OH(aq) level? Is it then in agreement with the ranges as suggested (Arakaki et al., 2013)?

8. Figures 1 and 2 need to be improved:
   - Captions: 'The chemical aqueous phase mechanism of glyoxal (oxalic acid)' is not very meaningful. At the minimum, specify that it is the oxidation (formation) pathways by chemical radical processes as represented in JAMOC.

   - Add the names of the species next to the structures in both figures.

   - Caption Figure 1: what do you mean by 'Glyaq donates all three species'?

   - Caption Figure 1: Are there any sources of aqueous phase glyoxal known at all?

   - in cloud water, glyoxal is predominantly present in its dihydrate form and should be represented as such in the figure. The mono hydrate may form if there is limited water available, and the unhydrated is likely not present at all (Ervens and Volkamer, 2010).

   -Figure 2: How is the phase partitioning of oxalic acid represented in JAMOC? Given that oxalate forms numerous salts and complexes in the condensed phase, the representation of the phase partitioning based on Henry's law is likely not appropriate.

   - Figure 2: Is the oxidation of oxalic acid by $NO_3$ ignored in the mechanism? If so, why?

9. Discussion and Figure 3: The extent is not clear to which the reduction of gas phase mixing ratios is due to uptake into the aqueous phase or due to chemical loss in either phase. I suggest showing total mixing ratios (i.e. gas + aqueous) which would give information on dissolution or net loss, respectively.

**Minor comments**

l. 17: 'liquid' should be replaced by 'aqueous'

l. 65: It is not clear which reaction is referred to here ('reaction with ozone with hydroxide'). Do you mean R2, i.e. the reaction of ozone with the superoxide anion radical ($O_2-$)?

l. 69: 'only outgassing depends on Henry's law constant' – I don't understand this. The standard equations used for the description of mass transfer, e.g. Eq-69 in (Sander, 1999), include the Henry's law constant which is needed to describe the deviation from equilibrium and thus the concentration gradient that drives the uptake or evaporation, respectively, of species.

l. 112: I disagree with the authors that 'little is known' about NO3 reactions. There are quite extensive data sets available for NO3 reactions with organic compounds in the aqueous phase, e.g., (Herrmann, 2003, 2015; Herrmann et al., 2010)

l. 127: Why was the rate constant of the dimers estimated as being twice as large as that of the monomer? Is there any reference for this?
According to the general kinetic theory, the number of collisions of molecules (which determines the rate constant) scales inversely proportional to molecular mass. Thus, the assumption of a higher rate constant for molecules with doubled mass seems counterintuitive. In addition, the rate constant will also depend on the number of available groups at which the radical attacks. However, since dimers (such as the glyoxal dimer, e.g. (Kua et al., 2008)) form cyclic structures, this trend does not justify a higher rate constant either.

l. 165: Several multiphase model studies have shown that the direct uptake from the gas phase, Fenton chemistry and H2O2 photolysis are the main OH(aq) sources in cloud droplets (Deguillaume et al., 2004; Ervens et al., 2003; Herrmann et al., 2005). Compared to these sources, what is the relative contribution of photolysis of organic compounds to OH sources in the aqueous phase?

l. 170: What refers the value 2.33 to? Do you mean 'enhanced compared to the gas phase photolysis rate'?

l. 183/184: Why is the relative humidity 70%? Shouldn't it be 100% in clouds?

l. 186: I think it should read 'a stable cloud droplet population'

l. 186: A liquid water content of $3 \times 10^{-1}$ g $L^{-1}$ does not seem an appropriate liquid water content as it would result in ~10000 droplets /cm3 (with diameter of 40 micrometers). Was indeed this LWC used in the model or is it a typo and should read 0.3 g/m3?

l. 222: 'reaction rates' should be 'rate constants' or 'rate coefficients'. A reaction rate is rate at which a concentration changes, i.e. $d[C]/dt = -k[C]$ whereas k is the rate constant and [C] is the reactant concentration (Seinfeld and Pandis, 2006).

l. 228: I am confused by this terminology. JAMOC stands for 'Jülich Aqueous phase mechanism of organic compounds', i.e. it is a chemical mechanism which is usually just a list of reactions and their parameters. Such a mechanism can then be implemented into a model that simulates, e.g. the formation of clouds and processing of chemical species?
Thus, rain-out is a process in a model, in which JAMOC comprises one module. Please clarify what you mean by 'model' and 'mechanism', respectively.

**Technical comments**

 l. 39: 'react' should be 'reacts'

l. 49: 'extend' should be 'extended'

l. 182: 'modells' should be 'models'

---

## Referee Comment (RC2) · Anonymous Referee #1 · 21 Nov 2020

Review of "Oxidation of low-molecular weight organic compounds in cloud
droplets: development of the JAMOC chemical mechanism in CAABA/MECCA (version 4.5.0gmdd)
Author(s): Simon Rosanka et al.

The authors present a new chemical mechanism (JAMOC), implemented in MECCA. JAMOC includes explicit oxidation steps of organic compounds in the aqueous phase of cloud droplets and thus exceeds previous aqueous phase chemistry mechanisms suitable for box, regional and global modeling. Such extensions are urgently needed as currently, particularly in global models, aqueous phase chemistry modules are largely limited to sulfur(IV) oxidation.
However, the current manuscript needs major clarifications and additions to make it a comprehensive and useful extension to previous multiphase model studies. The choice of the newly added reactions is not always evident and also the discussion of the example results in Figure 3 is misleading. In addition, at several places, terminology is confusing or inaccurate. Also previous literature on atmospheric multiphase modeling should be properly discussed. Overall, while I think that this reaction mechanism could possibly become a useful addition to the currently used ones, my comments below need to be carefully addressed prior to possible recommendation for publication.

**Major comments**

1. Oligomerization has been discussed to be only relevant in the aqueous phase of aerosol particles where organic concentrations may be higher than in cloud droplets (Ervens et al., 2015; Perri et al., 2009; Tan et al., 2009). Thus, it is not clear why they are included in the mechanism. While aqueous phase reactions might also occur in the aqueous phase of aerosol particles, such an extension would also need to include adjustments of Henry's law constants and kinetic reaction rate constants for high ionic strength. Thus, this seems out of the scope of the current study.

2. The authors discuss at several places the role of gem-diols and the need of including their phase partitioning between gas and aqueous phases. Is there any indication of the relevance of such processes? Given the very small concentration of water in the gas phase, the stability of gem-diols in the gas phase is likely very small. I assume that their gas phase fraction is likely negligible. Unless the authors can provide literature or estimates on their Henry's law constants that show the opposite, I would not identify the inclusion of their partitioning into atmospheric chemical mechanisms as one of the main gaps in current mechanism developments. In fact, the hydrated glyoxylic acid has been shown to be of such low volatility that it can be involved in new particle formation (Liu et al., 2017)
   - Please show the estimated Henry's law constants in Table 1.
   - Add references that indicate their potential relevance in the gas phase.

3. At several places in the manuscript, it is not clear whether the authors refer to predictions of aqueous phase rates and budgets or to both phases. For example,
   - l. 59: R1 is certainly not a major sink in the atmosphere, but only in the aqueous phase
   - l. 190: is the explicit oxidation of OVOCs only added in the aqueous or also in the gas phase?

4. At several places in the manuscript, misleading or wrong terminology is used, e.g.
   - l. 38: 'nitrogen trioxide' is usually referred to as 'nitrate radical'
   - l. 50: 'simulating hydration and dehydration explicitly' implies that the hydration reaction and dehydration reactions are implemented. However, it seems that only the equilibria are included, separately from the gas/aqueous phase partitioning described by Henry's law.

- l. 134: Do you mean 'recombination of alkyl radicals', i.e. the self-reaction of two radicals?

5.  The model studies are performed for a period of 5 days. However, the typical lifetime of a cloud droplet is on the order of 30 min or less (i.e. the time a droplet spends between cloud base and top in up- and downdraft regimes). You should at least mention that the model simulations are highly idealized and should be regarded as a sensitivity study rather than a realistic scenario.

6.  The authors imply that the additional OVOC oxidation in the aqueous leads to the significant decrease in predicted OH concentration. However, the main reason why significantly less OH is observed in the presence of clouds is the decrease in its formation rate in the gas phase from the reaction of $NO + HO_2$. These reactants are separated due to their significantly different water-solubility, e.g. (Jacob, 1986; Lelieveld and Crutzen, 1990). Consequently, the lower $HO_2$ and higher NO levels in the presence of clouds are not due to the different water solubilities of OH and $HO_2$ but because of the differences in the gas phase photochemical cycles of $HO_x$ and $NO_x$. This should be discussed and analyzed in Section 3.

7.  Given that currently only a small fraction (~ 15%) of organics in cloud water can be identified on a molecular basis, (e.g., Herckes et al., 2013), implies that also even the most detailed chemical aqueous phase mechanism is likely largely incomplete in terms of organic species. Thus, also the predicted OH(aq) concentration is likely biased high as by far not all sinks are included. The idea of the general scavenging rate constant as suggested by Arakaki et (2008) is that it can applied to parameterize the loss of OH(aq). Thus, it would seem a reasonable 'short-cut' to implement water-soluble organic carbon (WSOC) mass as an additional 'species' in the mechanism that reacts with OH(aq) to account for missing OH(aq) sources. As the products may be in many cases other WSOC compounds, this reaction could be implemented as WSOC + OH → WSOC + $HO_2$ (k = 3.8e8 M-1 s-1). How would the implementation of this reaction change the results in general, and in particular the OH(aq) level? Is it then in agreement with the ranges as suggested (Arakaki et al., 2013)?

8.  Figures 1 and 2 need to be improved:
    - Captions: 'The chemical aqueous phase mechanism of glyoxal (oxalic acid)' is not very meaningful. At the minimum, specify that it is the oxidation (formation) pathways by chemical radical processes as represented in JAMOC.

    - Add the names of the species next to the structures in both figures.

    - Caption Figure 1: what do you mean by 'Glyaq donates all three species'?

    - Caption Figure 1: Are there any sources of aqueous phase glyoxal known at all?

    - in cloud water, glyoxal is predominantly present in its dihydrate form and should be represented as such in the figure. The mono hydrate may form if there is limited water available, and the unhydrated is likely not present at all (Ervens and Volkamer, 2010).

    -Figure 2: How is the phase partitioning of oxalic acid represented in JAMOC? Given that oxalate forms numerous salts and complexes in the condensed phase, the representation of the phase partitioning based on Henry's law is likely not appropriate.

    - Figure 2: Is the oxidation of oxalic acid by $NO_3$ ignored in the mechanism? If so, why?

9. Discussion and Figure 3: The extent is not clear to which the reduction of gas phase mixing ratios is due to uptake into the aqueous phase or due to chemical loss in either phase. I suggest showing total mixing ratios (i.e. gas + aqueous) which would give information on dissolution or net loss, respectively.

**Minor comments**

l. 17: 'liquid' should be replaced by 'aqueous'

l. 65: It is not clear which reaction is referred to here ('reaction with ozone with hydroxide'). Do you mean R2, i.e. the reaction of ozone with the superoxide anion radical ($O_2-$)?

l. 69: 'only outgassing depends on Henry's law constant' – I don't understand this. The standard equations used for the description of mass transfer, e.g. Eq-69 in (Sander, 1999), include the Henry's law constant which is needed to describe the deviation from equilibrium and thus the concentration gradient that drives the uptake or evaporation, respectively, of species.

l. 112: I disagree with the authors that 'little is known' about NO3 reactions. There are quite extensive data sets available for NO3 reactions with organic compounds in the aqueous phase, e.g., (Herrmann, 2003, 2015; Herrmann et al., 2010)

l. 127: Why was the rate constant of the dimers estimated as being twice as large as that of the monomer? Is there any reference for this?
According to the general kinetic theory, the number of collisions of molecules (which determines the rate constant) scales inversely proportional to molecular mass. Thus, the assumption of a higher rate constant for molecules with doubled mass seems counterintuitive. In addition, the rate constant will also depend on the number of available groups at which the radical attacks. However, since dimers (such as the glyoxal dimer, e.g. (Kua et al., 2008)) form cyclic structures, this trend does not justify a higher rate constant either.

l. 165: Several multiphase model studies have shown that the direct uptake from the gas phase, Fenton chemistry and H2O2 photolysis are the main OH(aq) sources in cloud droplets (Deguillaume et al., 2004; Ervens et al., 2003; Herrmann et al., 2005). Compared to these sources, what is the relative contribution of photolysis of organic compounds to OH sources in the aqueous phase?

l. 170: What refers the value 2.33 to? Do you mean 'enhanced compared to the gas phase photolysis rate'?

l. 183/184: Why is the relative humidity 70%? Shouldn't it be 100% in clouds?

l. 186: I think it should read 'a stable cloud droplet population'

l. 186: A liquid water content of $3 \times 10^{-1}$ g $L^{-1}$ does not seem an appropriate liquid water content as it would result in ~10000 droplets /cm3 (with diameter of 40 micrometers). Was indeed this LWC used in the model or is it a typo and should read 0.3 g/m3?

l. 222: 'reaction rates' should be 'rate constants' or 'rate coefficients'. A reaction rate is rate at which a concentration changes, i.e. d[C]/dt = - k[C] whereas k is the rate constant and [C] is the reactant concentration (Seinfeld and Pandis, 2006).

l. 228: I am confused by this terminology. JAMOC stands for 'Jülich Aqueous phase mechanism of organic compounds', i.e. it is a chemical mechanism which is usually just a list of reactions and their parameters. Such a mechanism can then be implemented into a model that simulates, e.g. the formation of clouds and processing of chemical species?
Thus, rain-out is a process in a model, in which JAMOC comprises one module. Please clarify what you mean by 'model' and 'mechanism', respectively.

**Technical comments**

l. 39: 'react' should be 'reacts'

l. 49: 'extend' should be 'extended'

l. 182: 'modells' should be 'models'

**References**

Arakaki, T., Anastasio, C., Kuroki, Y., Nakajima, H., Okada, K., Kotani, Y., Handa, D., Azechi, S., Kimura, T., Tsuhako, A. and Miyagi, Y.: A general scavenging rate constant for reaction of hydroxyl radical with organic carbon in atmospheric waters, Environ. Sci. Technol., 47(15), 8196–8203, doi:10.1021/es401927b, 2013.

Deguillaume, L., Leriche, M., Monod, A. and Chaumerliac, N.: The role of transition metal ions on $HO_x$ radicals in clouds: a numerical evaluation of its impact on multiphase chemistry, Atmos. Chem. Phys., 4, 95–110, 2004.

Ervens, B. and Volkamer, R.: Glyoxal processing by aerosol multiphase chemistry: towards a kinetic modeling framework of secondary organic aerosol formation in aqueous particles, Atmos. Chem. Phys., 10, 8219–8244, 2010.

Ervens, B., George, C., Williams, J. E., Buxton, G. V, Salmon, G. A., Bydder, M., Wilkinson, F., Dentener, F., Mirabel, P., Wolke, R. and Herrmann, H.: CAPRAM2.4 (MODAC mechanism): An extended and condensed tropospheric aqueous phase mechanism and its application, J. Geophys. Res., 108(D14), 4426, doi:doi: 10.1029/2002JD002202, 2003.

Ervens, B., Renard, P., Tlili, S., Ravier, S., Clément, J. L. and Monod, A.: Aqueous-phase oligomerization of methyl vinyl ketone through photooxidation – Part 2: Development of the chemical mechanism and atmospheric implications, Atmos. Chem. Phys., 15(16), 9109–9127, doi:10.5194/acp-15-9109-2015, 2015.

Herckes, P., Valsaraj, K. T. and Collett, J. L.: A review of observations of organic matter in fogs and clouds: origin, processing and fate, Atmos Res, 132–133, doi:10.1016/j.atmosres.2013.06.005, 2013.

Herrmann, H.: Kinetics of aqueous phase reactions relevant for atmospheric chemistry, Chem. Rev., 103(12), 4691–4716, 2003.

Herrmann, H.: Tropospheric aqueous-phase chemistry: kinetics, mechanisms, and its coupling to a

changing gas phase, Chem Rev, 115, doi:10.1021/cr500447k, 2015.

Herrmann, H., Tilgner, A., Barzaghi, P., Majdik, Z., Gligorovski, S., Poulain, L. and Monod, A.: Towards a more detailed description of tropospheric aqueous phase organic chemistry: CAPRAM 3.0, Atmos. Env., 39, 4351–4363, doi:10.1016/j.atmosenv.2005.02.016, 2005.

Herrmann, H., Hoffmann, D., Schaefer, T., Bräuer, P. and Tilgner, A.: Tropospheric Aqueous-Phase Free-Radical Chemistry: Radical Sources, Spectra, Reaction Kinetics and Prediction Tools, ChemPhysChem, 11(18), 3796–3822, doi:10.1002/cphc.201000533, 2010.

Jacob, D. J.: Chemistry of OH in Remote Clouds and its Role in the Production of Formic Acid and Peroxymonosulfate, J. Geophys. Res. - Atmos., 91, 9807–9826 [online] Available from: http://onlinelibrary.wiley.com/doi/10.1029/JD091iD09p09807/pdf, 1986.

Kua, J., Hanley, S. W. and DeHaan, D. O.: Thermodynamics and kinetics of glyoxal dimer formation: A computational study, J. Phys. Chem. A, 112(1), 66–72, 2008.

Lelieveld, J. and Crutzen, P. J.: Influences of cloud photochemical processes on tropospheric ozone, Nature, 343, 227–233, doi:10.1007/BF00048075, 1990.

Liu, L., Zhang, X., Li, Z., Zhang, Y. and Ge, M.: Gas-phase hydration of glyoxylic acid: Kinetics and atmospheric implications, Chemosphere, 186, 430–437, doi:https://doi.org/10.1016/j.chemosphere.2017.08.007, 2017.

Perri, M. J., Seitzinger, S. and Turpin, B. J.: Secondary organic aerosol production from aqueous photooxidation of glycolaldehyde: Laboratory experiments, Atmos. Environ., 43(8), 1487–1497, doi:http://dx.doi.org/10.1016/j.atmosenv.2008.11.037, 2009.

Sander, R.: Modeling Atmospheric Chemistry: Interactions between Gas-Phase Species and Liquid Cloud/Aerosol Particle, Surv. Geophys., 20, 1–31, 1999.

Seinfeld, J. H. and Pandis, S. N.: Atmospheric Chemistry and Physics - From air pollution to climate change, 2nd ed., edited by I. John Wiley and Sons, John Wiley & Sons, Inc., Hoboken, New Jersey., 2006.

Tan, Y., Perri, M. J., Seitzinger, S. P. and Turpin, B. J.: Effects of Precursor Concentration and Acidic Sulfate in Aqueous Glyoxal-OH Radical Oxidation and Implications for Secondary Organic Aerosol, Environ. Sci. Technol., 43(21), 8105–8112, doi:10.1021/es901742f, 2009.

---

## Referee Comment (RC3) · Anonymous Referee #2 · 27 Nov 2020

The paper "Oxidation of low-molecular weight organic compounds in cloud droplets: development of the JAMOC chemical mechanism in CAABA/MECCA (version 4.5.0gmdd)" by Rosanka et al. presents JAMOC, a cloud chemical mechanism. The paper fits within the scope of GMD and I recommend publication once the authors address a couple of points and provide additional information.

General Comments

I have a little difficulty in understanding how JAMOC fits within the whole

[Figure]

MECCA/CAABA/MESSy/EMAC ecosystem and while I am sure this has been described previously in the corresponding papers, I think it should be mentioned repeated here. A few points come to mind:

1. As I understand it, the JAMOC module is part of MECCA and MECCA itself is a module that can be used in the CAABA box-model and/or in the EMAC global model. In this case, it is perhaps better to remove CAABA from the title (and elsewhere in the text)?

2. If I am not mistaken, MECCA already includes an aqueous-phase chemical mechanism. Is JAMOC an upgrade/extension to it or is it supposed to replace it or run alongside it? This relationship should be clarified, and it should be explained, for example, why it is necessary to create a new module and not simply add reactions to the pre-existing aqueous-phase module.

3. On page 3 the authors say: "The inorganic chemistry for the proposed mechanism is very similar to the inorganic chemistry of the standard aqueous-phase mechanism used in EMAC (Tost et al., 2007; Jockel et al., 2016) and reactions included in MECCA (e.g. Fenton chemistry), which are not yet implemented in EMAC." This is a bit confusing, as it implies that the reactions in JAMOC are already in MECCA and some, but not all, are already in EMAC. Howeverm it does not say which ones, so it is difficult to understand how much of a change has been made. This also suggests that JAMOC is meant to replace the current aqueous-phase mechanism in MECCA and/or EMAC. As per my previous point, the relationship between the different parts of the system needs to be clarified.

The other major comment I have is about CLEPS. The authors describe JAMOC as derived from CLEPS but they don't say how this was done. Were the reactions "hand-picked" from CLEPS (if so on which basis?) or was some reduction procedure applied? It is important that the process is described and the rationale behind certain choices is explained.

I also think it would be more useful and, more informative perhaps, to compare the output of "CAABA with JAMOC" to the output of CLEPS, rather than to the output of "CAABA without JAMOC" (Figure 3 and related discussion). This would allow a better understanding of the accuracy of the reduction procedure, and how much information (if any) is lost when the more explicit mechanism CLEPS is condensed into the smaller mechanism JAMOC.

Minor Comments

line48: "only a selection of species containing up to four carbon atoms react within the aqueous-phase". Can you explain why this choice was made? I understand one of the reasons is to keep the size of the mechanism relatively small, but why only up to C4 species react and up to C10 species undergo phase transfer? Is the reaction of molecules with high carbon number too slow to matter? Do you actually need to transfer C5-C10 into the aqueous-phase if they don't react and you need to keep the mechanism small?

line 71: can you clarify the difference between apparent and intrinsic Henry's law constant?

line 86: "Pseudo-first order rate constants for the hydration and dehydration are mainly obtained from the literature". This implies that some were obtained or estimated in another way, please clarify.

line 141: "In all cases, branching ratios are rescaled to 100%." Can you explain this point better?

line 149: how much faster is R11 with respect to R12?

line 182: correct "modells"

figure 3: the first panel should be sum of OVOC rather than sum of VOCs
* * *
[Figure]

2020.

---

## Author Comment (AC1) · 5 Mar 2021

**Reply to comments of Anonymous Referee #1**

The authors present a new chemical mechanism (JAMOC), implemented in MECCA. JAMOC includes explicit oxidation steps of organic compounds in the aqueous phase of cloud droplets and thus exceeds previous aqueous phase chemistry mechanisms suitable for box, regional and global modeling. Such extensions are urgently needed as currently, particularly in global models, aqueous phase chemistry modules are largely limited to sulfur(IV) oxidation. However, the current manuscript needs major clarifications and additions to make it a comprehensive and useful extension to previous multiphase model studies. The choice of the newly added reactions is not always evident and also the discussion of the example results in Figure 3 is misleading. In addition, at several places, terminology is confusing or inaccurate. Also previous literature on atmospheric multiphase modeling should be properly discussed. Overall, while I think that this reaction mechanism could possibly become a useful addition to the currently used ones, my comments below need to be carefully addressed prior to possible recommendation for publication.

*Thank you very much for the helpful comments and seeing the value of our work to the community. Please find in black the original comments and in red our replies. We significantly extended the introduction, in order to meet your request to discuss previous literature more extensively.*

**Major comments**

1. Oligomerization has been discussed to be only relevant in the aqueous phase of aerosol particles where organic concentrations may be higher than in cloud droplets (Ervens et al., 2015; Perri et al., 2009; Tan et al., 2009). Thus, it is not clear why they are included in the mechanism. While aqueous phase reactions might also occur in the aqueous phase of aerosol particles, such an extension would also need to include adjustments of Henry's law constants and kinetic reaction rate constants for high ionic strength. Thus, this seems out of the scope of the current study.

   *We agree with the reviewer that the importance of oligomerization is higher for typical concentrations found in aerosol water (see Tan et al., 2009). At the same time, earlier modelling studies demonstrated that the secondary organic aerosol formation from cloud processing is globally important (Lin et al., 2012). In a future study, we plan to apply JAMOC, including the necessary adjustments, to the aqueous phase of aerosol particles. Thus, we prefer to keep the representation of oligomerization in JAMOC in order to represent clouds as SOA sources.*

2. The authors discuss at several places the role of gem-diols and the need of including their phase partitioning between gas and aqueous phases. Is there any indication of the relevance of such processes? Given the very small concentration of water in the gas phase, the stability of gem-diols in the gas phase is likely very small. I assume that their gas phase fraction is likely negligible. Unless the authors can provide literature or estimates on their Henry's law constants that show the opposite, I would not identify the inclusion of their partitioning into atmospheric chemical mechanisms as one of the main gaps in current mechanism developments. In fact, the hydrated glyoxylic acid has been shown to be of such low volatility that it can be involved in new particle formation (Liu et al., 2017)

   - Please show the estimated Henry's law constants in Table 1.

   - Add references that indicate their potential relevance in the gas phase.

   *Overall, the dehydration of many gem-diols is slower than the typical lifetime of cloud droplets. Concerning the stability of gem-diols in the gas phase, Kumar et al. (2017) calculated that for the methanediol the shortest lifetime against decomposition by HCOOH-catalysis is larger than 1E9 s. We agree with the reviewer that the inclusion of the gem-diol partitioning is not one of the main gaps in current mechanism development. The reason why we discuss this aspect at multiple places is the fact that this aspect clearly separates JAMOC from CLEPS. JAMOC is designed to be used within global models and the transfer of gem-diols into the gas-phase, once the cloud droplet evaporates, is only possible if their partitioning is explicitly represented within JAMOC. Investigating the importance of their gas-phase oxidation is outside the scope of the manuscript. However, we are currently investigating this process on a global scale and*

plan further publications with this focus. Therefore, we want to keep the gem-diol partitioning and oxidation in the developed mechanism. In the revised manuscript, we added an explanation of this mechanistic and added a new table summarising all estimated Henry's law constants.

3. At several places in the manuscript, it is not clear whether the authors refer to predictions of aqueous phase rates and budgets or to both phases. For example,

   - l. 59: R1 is certainly not a major sink in the atmosphere, but only in the aqueous phase

   This is indeed correct. In the revised manuscript, it is now explicitly stated that R1 is only the major sink in the aqueous-phase.

   - l. 190: is the explicit oxidation of OVOCs only added in the aqueous or also in the gas phase?

   Within the development of JAMOC, only the oxidations of the gem-diols and oxalic acid are added to the gas-phase mechanism. For all other OVOCs, the gas-phase oxidation is represented by the Mainz Organic Mechanism (MOM, Sander et al., 2019). In the revised manuscript, this now reads as "With the explicit oxidation of many OVOCs in the aqueous-phase [...]".

4. At several places in the manuscript, misleading or wrong terminology is used, e.g.

   - l. 38: 'nitrogen trioxide' is usually referred to as 'nitrate radical'

   We changed it to nitrate radical.

   - l. 50: 'simulating hydration and dehydration explicitly' implies that the hydration reaction and dehydration reactions are implemented. However, it seems that only the equilibria are included, separately from the gas/aqueous phase partitioning described by Henry's law.

   Even though we list only equilibrium constants in Table 1, the model does indeed calculate hydration and dehydration explicitly. The forward and backward rate constants can be found in the supplement in the file aqueous.eqn.

   - l. 134: Do you mean 'recombination of alkyl radicals', i.e. the self-reaction of two radicals?

   Exactly. For clarification this now reads "Thus, oligomers formed from the self- and cross-reactions of alkyl radicals are not considered in JAMOC." in the revised manuscript.

5. The model studies are performed for a period of 5 days. However, the typical lifetime of a cloud droplet is on the order of 30 min or less (i.e. the time a droplet spends between cloud base and top in up-and downdraft regimes). You should at least mention that the model simulations are highly idealized and should be regarded as a sensitivity study rather than a realistic scenario.

   Indeed, cloud droplets are short lived and the simulations presented in this study are highly idealised. Exactly for this reason, we present a "realistic" cloud event box-model study (i.e. cloud droplet lifetime of 1 hour) in our companion paper by Rosanka et al. (2020). Here, we used the highly idealised CAABA simulations to perform a sensitivity study using JAMOC. We now state the highly idealised nature of this simulations and point to Rosanka et al. (2020) for the realistic cloud scenario simulations using CAABA and EMAC.

6. The authors imply that the additional OVOC oxidation in the aqueous leads to the significant decrease in predicted OH concentration. However, the main reason why significantly less OH is observed in the presence of clouds is the decrease in its formation rate in the gas phase from the reaction of NO + HO2. These reactants are separated due to their significantly different water-solubility, e.g. (Jacob, 1986; Lelieveld and Crutzen, 1990). Consequently, the lower HO2 and higher NO levels in the presence of clouds are not due to the different water solubilities of OH and HO2 but because of the differences in the gas phase photochemical cycles of HOx and NOx. This should be discussed and analyzed in Section 3.

   Thank you for pointing this out. We reformulated the section significantly.

7. Given that currently only a small fraction ($\sim$15%) of organics in cloud water can be identified on a molecular basis, (e.g. Herckes et al., 2013), implies that also even the most detailed chemical aqueous phase mechanism is likely largely incomplete in terms of organic species. Thus, also the predicted OH(aq) concentration is likely biased high as by far not all sinks are included. The idea of the general scavenging rate constant as suggested by Arakaki et (2008) is that

it can applied to parameterize the loss of OH(aq). Thus, it would seem a reasonable 'short-cut'to implement water-soluble organic carbon (WSOC) mass as an additional 'species' in the mechanism that reacts with OH(aq) to account for missing OH(aq) sources. As the products may be in many cases other WSOC compounds, this reaction could be implemented as WSOC + OH → WSOC + HO2 (k = 3.8e8 M-1 s-1). How would the implementation of this reaction change the results in general, and in particular the OH(aq) level? Is it then in agreement with the ranges as suggested (Arakaki et al., 2013)?

In general, JAMOC is targeted for global model applications and focuses on the explicit oxidation kinetics of known compounds. Even though most of the products from WSOC oxidation are expected to be other WSOCs, the real products are unknown. If the WSOC oxidation was implemented using the suggested equation (WSOC + OH → WSOC + $HO_2$), the concentration of each species in the WSOC group would stay the same. Within a global model, all WSOC are transferred into the gas-phase, when the cloud droplet evaporates. Here, the same species, which was artificially oxidised in the aqueous-phase, would undergo an additional oxidation. In addition, we estimate that each $HO_2$ formed from WSOC oxidation would roughly return one OH. Therefore, this approach is not suited for global model applications.

8. Figures 1 and 2 need to be improved:

- Captions: 'The chemical aqueous phase mechanism of glyoxal (oxalic acid)' is not very meaningful. At the minimum, specify that it is the oxidation (formation) pathways by chemical radical processes as represented in JAMOC.

In the revised version of the manuscript, both captions are updated following your suggestions.

- Add the names of the species next to the structures in both figures.

Done.

- Caption Figure 1: what do you mean by 'Glyaq donates all three species'?

In JAMOC, the oligomerisation of glyoxal is implemented such that the mono-hydrate reacts with glyoxal, the glyoxal mono-hydrate, and the glyoxal dihydrate. For visual simplicity, we represent all three species by $Gly_{aq}$. In the caption, it now states "Here, $Gly_{aq}$ denotes all three forms of glyoxal (glyoxal, the glyoxal mono-hydrate, and the glyoxal dihydrate), which is consistent with the kinetic data published by Ervens and Volkamer (2010)".

- Caption Figure 1: Are there any sources of aqueous phase glyoxal known at all?

Glyoxal and the glyoxal mono-hydrate are formed during the $HO_2$ elimination of the peroxyl radicals formed from the oxidation of glycolaldehyde and the glycolaldehyde mono-hydrate. In the revised version, the caption of Fig. 1 now includes this additional information.

- in cloud water, glyoxal is predominantly present in its dihydrate form and should be represented as such in the figure. The mono hydrate may form if there is limited water available, and the unhydrated is likely not present at all (Ervens and Volkamer, 2010).

We agree that with the reviewer that glyoxal is predominantly present as a dihydrate in cloud droplets. However, within JAMOC all hydration steps are represented and Fig. 1 is supposed to show JAMOC's representation of glyoxal oxidation (as stated in the updated caption). Additionally, we plan to apply JAMOC (with the right adjustments) to aerosols with lower water availability.

- Figure 2: How is the phase partitioning of oxalic acid represented in JAMOC? Given that oxalate forms numerous salts and complexes in the condensed phase, the representation of the phase partitioning based on Henry's law is likely not appropriate.

We have to admit that we do not include the formation of metal complexes in the mechanism yet. However, we plan to implement this for iron (Fe) soon in a follow up publication. Within JAMOC, the phase partitioning of oxalic acid is represented based on the Henry's law. This is necessary, since if the phase transfer is not represented by the Henry's law, oxalic acid will not be transferred into the gas-phase by EMAC when the cloud droplet evaporates. Additionally, we plan to apply JAMOC to aerosols (with the necessary adjustments). Due to the lower pH in aerosol water, the representation by the Henry's law is appropriate.

- Figure 2: Is the oxidation of oxalic acid by NO3 ignored in the mechanism? If so, why?

For the $NO_3$ oxidation of oxalic acid, no rate constant is known from the literature. Opposite to the oxidation by OH, no structure activity relationship (SAR) is available for the H-abstraction implemented for the $NO_3$ oxidation within JAMOC. Therefore, within CLEPS, similarity criteria are applied if no rate constant is known. However, this is not available for oxalic acid. Therefore, JAMOC does not include the oxidation of oxalic acid by $NO_3$. In the revised manuscript, we added an explanation on the rate constant and branching ratios for the oxidation by OH and $NO_3$ to the respective section.

9. Discussion and Figure 3: The extent is not clear to which the reduction of gas phase mixing ratios is due to uptake into the aqueous phase or due to chemical loss in either phase. I suggest showing total mixing ratios (i.e. gas + aqueous) which would give information on dissolution or net loss, respectively.

We updated Fig. 3 according to your comment. We now show total mixing ratios (i.e. gas + aqueous) for both mechanisms used. In addition, we added an extra line displaying the aqueous-phase mixing ratio of $\sum$OVOCs, methanol, glycolaldehyde, and methylglyoxal for the simulation using JAMOC. The caption was revised accordingly. Following a suggestion of the first referee of our companion paper (Rosanka et al., 2020), we moved the definition of $\sum$OVOCs from the caption of Fig. 3 into the newly created Appendix A1.

**Minor comments**

l. 17: 'liquid' should be replaced by 'aqueous'

We changed it accordingly.

l. 65: It is not clear which reaction is referred to here ('reaction with ozone with hydroxide'). Do you mean R2, i.e. the reaction of ozone with the superoxide anion radical (O2-)?

In its original version, this statement was supposed to imply that the complete mechanism proposed by Staehelin et al. (1984) and Staehelin and Hoigné (1985) is used in JAMOC. However, we agree that this statement might be confusing. Therefore, we updated this paragraph in the revised version of the manuscript.

l. 69: 'only outgassing depends on Henry's law constant' –I don't understand this. The standard equations used for the description of mass transfer, e.g. Eq-69 in (Sander, 1999), include the Henry's law constant which is needed to describe the deviation from equilibrium and thus the concentration gradient that drives the uptake or evaporation, respectively, of species.

We removed this statement in the revised manuscript.

l. 112: I disagree with the authors that 'little is known' about NO3 reactions. There are quite extensive data sets available for NO3 reactions with organic compounds in the aqueous phase, e.g., (Herrmann, 2003; Herrmann et al., 2015, 2010)

After carefully reevaluating the current literature, we agree with the referee's comment and removed this statement from the revised manuscript.

l. 127: Why was the rate constant of the dimers estimated as being twice as large as that of the monomer? Is there any reference for this? According to the general kinetic theory, the number of collisions of molecules (which determines the rate constant) scales inversely proportional to molecular mass. Thus, the assumption of a higher rate constant for molecules with doubled mass seems counterintuitive. In addition, the rate constant will also depend on the number of available groups at which the radical attacks. However, since dimers (such as the glyoxal dimer, e.g. (Kua et al., 2008)) form cyclic structures, this trend does not justify a higher rate constant either.

Kua et al. (2008) presents theoretical calculations only of the kinetics of dimerization and not kinetics in the reaction with OH. We estimated the rate constant to be twice as large as the one of the monomer, because the dimers have almost double the number of abstractable H-atoms than the monomers. This reasoning is now included in the respective section.

l. 165: Several multiphase model studies have shown that the direct uptake from the gas phase,

Fenton chemistry and H2O2 photolysis are the main OH(aq) sources in cloud droplets (Deguillaume et al., 2004; Ervens et al., 2003; Herrmann et al., 2005). Compared to these sources, what is the relative contribution of photolysis of organic compounds to OH sources in the aqueous phase?

In Rosanka et al. (2020), we apply JAMOC globally by using EMAC and provide the first global in-cloud OH budget (see Table 2 in the initial submission of Rosanka et al., 2020). Here, the formation of OH from the photolysis of OVOCs containing one carbon atom is estimated to be more than four times higher than the photolysis of $H_2O_2$. Additionally, we estimate that the photolysis of OVOCs containing more than one carbon atom to contribute about one third when compared to the photolysis of $H_2O_2$. A further elaboration of this is added to the revised manuscript.

l. 170: What refers the value 2.33 to? Do you mean 'enhanced compared to the gas phase photolysis rate'?

Indeed. In general, gas-phase photolysis rates are available. In order to account for scattering effects within cloud droplets, we apply an enhancement factor of 2.33 to the gas-phase photolysis rates within JAMOC. In order to clarify this, we adjusted the text to: "In order to account for scattering effects within cloud droplets, an enhancement factor of 2.33, the same as used in EMAC's standard aqueous-phase mechanism for the photolysis of $H_2O_2$, is applied to each gas-phase photolysis rate."

l. 183/184: Why is the relative humidity 70%? Shouldn't it be 100% in clouds?

For the cloud scenario simulated, we increased the relative humidity to 100 %. We performed both simulations again and updated Fig. 3.

l. 186: I think it should read 'a stable cloud droplet population'

We changed this in the revised manuscript.

l. 186: A liquid water content of 3 x 10-1g L-1 does not seem an appropriate liquid water content as it would result in 10000 droplets /cm3 (with diameter of 40 micrometers). Was indeed this LWC used in the model or is it a typo and should read 0.3 g/m3?

Thank you for pointing this out. We checked our simulation setup and the LWC we used is indeed 0.3 g/m3 and the reported value in the manuscript is a typo. We changed it in the revised manuscript.

l. 222: 'reaction rates' should be 'rate constants' or 'rate coefficients'. A reaction rate is rate at which a concentration changes, i.e.d[C]/dt = -k[C] whereas k is the rate constant and [C] is the reactant concentration (Seinfeld and Pandis, 2006).

We agree with the reviewer. We changed it accordingly.

l. 228: I am confused by this terminology. JAMOC stands for 'Jülich Aqueous phase mechanism of organic compounds', i.e. it is a chemical mechanism which is usually just a list of reactions and their parameters. Such a mechanism can then be implemented into a model that simulates, e.g. the formation of clouds and processing of chemical species? Thus, rain-out is a process in a model, in which JAMOC comprises one module. Please clarify what you mean by 'model' and 'mechanism', respectively.

By representing the phase-transfer of organic species, which are not explicitly oxidised in JAMOC, allows to represent their rain-out when using global models. We agree with the reviewer that the current wording is confusing. We updated this statment and it now reads: "Phase-transfer of soluble VOCs into cloud droplets is considered in JAMOC even when their oxidation is not explicitly represented (see Sect. 2.2). This allows their removal from the atmosphere by rain-out when JAMOC is connected to a global model (e.g. using EMAC, see Rosanka et al., 2020)."

**Technical comments**

l. 39: 'react' should be 'reacts'

l. 49: 'extend' should be 'extended'

l. 182: 'modells' should be 'models'

Thank you for spotting these three mistakes. We adjusted them.

**References**

Arakaki, T., Anastasio, C., Kuroki, Y., Nakajima, H., Okada, K., Kotani, Y., Handa, D., Azechi, S., Kimura, T., Tsuhako, A., and Miyagi, Y.: A General Scavenging Rate Constant for Reaction of Hydroxyl Radical with Organic Carbon in Atmospheric Waters, Environmental Science & Technology, 47, 8196–8203, https://doi.org/10.1021/es401927b, pMID: 23822860, 2013.

Deguillaume, L., Leriche, M., Monod, A., and Chaumerliac, N.: The role of transition metal ions on $HO_x$ radicals in clouds: a numerical evaluation of its impact on multiphase chemistry, Atmospheric Chemistry and Physics, 4, 95–110, https://doi.org/10.5194/acp-4-95-2004, 2004.

Ervens, B. and Volkamer, R.: Glyoxal processing by aerosol multiphase chemistry: towards a kinetic modeling framework of secondary organic aerosol formation in aqueous particles, Atmospheric Chemistry and Physics, 10, 8219–8244, https://doi.org/10.5194/acp-10-8219-2010, 2010.

Ervens, B., George, C., Williams, J. E., Buxton, G. V., Salmon, G. A., Bydder, M., Wilkinson, F., Dentener, F., Mirabel, P., Wolke, R., and Herrmann, H.: CAPRAM 2.4 (MODAC mechanism): An extended and condensed tropospheric aqueous phase mechanism and its application, Journal of Geophysical Research: Atmospheres, 108, https://doi.org/https://doi.org/10.1029/2002JD002202, 2003.

Ervens, B., Renard, P., Tlili, S., Ravier, S., Clément, J.-L., and Monod, A.: Aqueous-phase oligomerization of methyl vinyl ketone through photooxidation – Part 2: Development of the chemical mechanism and atmospheric implications, Atmospheric Chemistry and Physics, 15, 9109–9127, https://doi.org/10.5194/acp-15-9109-2015, 2015.

Herckes, P., Valsaraj, K., and Collett, J.: A review of observations of organic matter in fogs and clouds: Origin, processing and fate, Atmospheric Research, 132-133, 434–449, https://doi.org/10.1016/j.atmosres.2013.06.005, 2013.

Herrmann, H.: Kinetics of Aqueous Phase Reactions Relevant for Atmospheric Chemistry, Chemical Reviews, 103, 4691–4716, https://doi.org/10.1021/cr020658q, pMID: 14664629, 2003.

Herrmann, H., Tilgner, A., Barzaghi, P., Majdik, Z., Gligorovski, S., Poulain, L., and Monod, A.: Towards a more detailed description of tropospheric aqueous phase organic chemistry: CAPRAM 3.0, Atmospheric Environment, 39, 4351 – 4363, https://doi.org/https://doi.org/10.1016/j.atmosenv.2005.02.016, URL http://www.sciencedirect.com/science/article/pii/S1352231005001937, fEBUKO and MODMEP: A Combined Study of Aerosol-Cloud Interaction by Field Experiments and Model Development, 2005.

Herrmann, H., Hoffmann, D., Schaefer, T., Bräuer, P., and Tilgner, A.: Tropospheric Aqueous-Phase Free-Radical Chemistry: Radical Sources, Spectra, Reaction Kinetics and Prediction Tools, ChemPhysChem, 11, 3796–3822, https://doi.org/https://doi.org/10.1002/cphc.201000533, 2010.

Herrmann, H., Schaefer, T., Tilgner, A., Styler, S. A., Weller, C., Teich, M., and Otto, T.: Tropospheric Aqueous-Phase Chemistry: Kinetics, Mechanisms, and Its Coupling to a Changing Gas Phase, Chemical Reviews, 115, 4259–4334, https://doi.org/10.1021/cr500447k, pMID: 25950643, 2015.

Jacob, D. J.: Chemistry of OH in remote clouds and its role in the production of formic acid and peroxymonosulfate, Journal of Geophysical Research: Atmospheres, 91, 9807–9826, https://doi.org/https://doi.org/10.1029/JD091iD09p09807, 1986.

Kua, J., Hanley, S. W., and De Haan, D. O.: Thermodynamics and Kinetics of Glyoxal Dimer Formation: A Computational Study, The Journal of Physical Chemistry A, 112, 66–72, https://doi.org/10.1021/jp076573g, pMID: 18067276, 2008.

Kumar, M., Anglada, J. M., and Francisco, J. S.: Role of Proton Tunneling and Metal-Free Organocatalysis in the Decomposition of Methanediol: A Theoretical Study, The Journal of Physical Chemistry A, 121, 4318–4325, https://doi.org/10.1021/acs.jpca.7b01864, pMID: 28513154, 2017.

Lelieveld, J. and Crutzen, P. J.: Influences of cloud photochemical processes on tropospheric ozone, Nature, 343, 227–233, https://doi.org/10.1038/343227a0, URL `https://doi.org/10.1038/343227a0`, 1990.

Lin, G., Penner, J. E., Sillman, S., Taraborrelli, D., and Lelieveld, J.: Global modeling of SOA formation from dicarbonyls, epoxides, organic nitrates and peroxides, Atmospheric Chemistry and Physics, 12, 4743–4774, https://doi.org/10.5194/acp-12-4743-2012, 2012.

Liu, L., Zhang, X., Li, Z., Zhang, Y., and Ge, M.: Gas-phase hydration of glyoxylic acid: Kinetics and atmospheric implications, Chemosphere, 186, 430 – 437, https://doi.org/https://doi.org/10.1016/j.chemosphere.2017.08.007, URL `http://www.sciencedirect.com/science/article/pii/S0045653517312298`, 2017.

Perri, M. J., Seitzinger, S., and Turpin, B. J.: Secondary organic aerosol production from aqueous photooxidation of glycolaldehyde: Laboratory experiments, Atmospheric Environment, 43, 1487 – 1497, https://doi.org/https://doi.org/10.1016/j.atmosenv.2008.11.037, URL `http://www.sciencedirect.com/science/article/pii/S1352231008011096`, 2009.

Rosanka, S., Sander, R., Franco, B., Wespes, C., Wahner, A., and Taraborrelli, D.: Oxidation of low-molecular weight organic compounds in cloud droplets: global impact on tropospheric oxidants, Atmos. Chem. Phys. Discuss., https://doi.org/10.5194/acp-2020-1041, 2020.

Sander, R.: Modeling Atmospheric Chemistry: Interactions between Gas-Phase Species and Liquid Cloud/Aerosol Particles, Surveys in Geophysics, 20, 1–31, https://doi.org/10.1023/A:1006501706704, URL `https://doi.org/10.1023/A:1006501706704`, 1999.

Sander, R., Baumgaertner, A., Cabrera-Perez, D., Frank, F., Gromov, S., Grooß, J.-U., Harder, H., Huijnen, V., Jöckel, P., Karydis, V. A., Niemeyer, K. E., Pozzer, A., Riede, H., Schultz, M. G., Taraborrelli, D., and Tauer, S.: The community atmospheric chemistry box model CAABA/MECCA-4.0, Geoscientific Model Development, 12, 1365–1385, https://doi.org/10.5194/gmd-12-1365-2019, 2019.

Seinfeld, J. H. and Pandis, S. N.: Atmospheric chemistry and physics – From air pollution to climate change, John Wiley & Sons, 2006.

Staehelin, J. and Hoigné, J.: Decomposition of ozone in water in the presence of organic solutes acting as promoters and inhibitors of radical chain reactions, Environmental Science & Technology, 19, 1206–1213, https://doi.org/10.1021/es00142a012, 1985.

Staehelin, J., Buehler, R. E., and Hoigné, J.: Ozone decomposition in water studied by pulse radiolysis. 2. Hydroxyl and hydrogen tetroxide (HO4) as chain intermediates, The Journal of Physical Chemistry, 88, 5999–6004, https://doi.org/10.1021/j150668a051, 1984.

Tan, Y., Perri, M. J., Seitzinger, S. P., and Turpin, B. J.: Effects of Precursor Concentration and Acidic Sulfate in Aqueous Glyoxal–OH Radical Oxidation and Implications for Secondary Organic Aerosol, Environmental Science & Technology, 43, 8105–8112, https://doi.org/10.1021/es901742f, pMID: 19924930, 2009.

---

## Author Comment (AC2) · 5 Mar 2021

**Reply to comments of Anonymous Referee #2**

The paper "Oxidation of low-molecular weight organic compounds in cloud droplets: development of the JAMOC chemical mechanism in CAABA/MECCA (version4.5.0gmdd)" by Rosanka et al. presents JAMOC, a cloud chemical mechanism. The paper fits within the scope of GMD and I recommend publication once the authors address a couple of points and provide additional information.

Thank you very much for the helpful comments and seeing the value of our work to the community. Please find in black the original comments and in red our replies.

**General Comments**

I have a little difficulty in understanding how JAMOC fits within the whole MECCA/CAABA/MESSy/EMAC ecosystem and while I am sure this has been described previously in the corresponding papers, I think it should be mentioned repeated here. A few points come to mind:

1. As I understand it, the JAMOC module is part of MECCA and MECCA itself is a module that can be used in the CAABA box-model and/or in the EMAC global model. In this case, it is perhaps better to remove CAABA from the title (and elsewhere in the text)?

   The reviewer correctly states that MECCA chemistry can be used inside the CAABA box model as well as inside a larger 3D model, e.g., EMAC. In this model description paper, however, we only evaluate MECCA chemistry in the CAABA box model. Therefore, we would like to keep the title as it is. The global impact of the newly implemented OVOC chemistry is analyzed with EMAC in a companion paper (Rosanka et al., 2020).

2. If I am not mistaken, MECCA already includes an aqueous-phase chemical mechanism. Is JAMOC an upgrade/extension to it or is it supposed to replace it or run alongside it? This relationship should be clarified, and it should be explained, for example, why it is necessary to create a new module and not simply add reactions to the preexisting aqueous-phase module.

   We fully agree with the reviewer that the relationship between the different model parts needs to be explained better. JAMOC is an addition to the aqueous-phase chemical mechanism, not a replacement. The main reason for the confusion is probably the modularity and flexibility of the MECCA system. Model users select an individual subset created from different components of the chemical mechanism. Thus, the new aqueous-phase OVOC chemistry from JAMOC can be switched on or off in MECCA as desired. This is now visualized in the new Figure 3 in the User Manual (included in the supplement). In the revised manuscript, it is now elaborated that JAMOC is considered an addition to MECCA's existing aqueous-phase chemical mechanism.

3. On page 3 the authors say: "The inorganic chemistry for the proposed mechanism is very similar to the inorganic chemistry of the standard aqueous-phase mechanism used in EMAC (Tost et al., 2007; Jöckel et al., 2016) and reactions included in MECCA (e.g. Fenton chemistry), which are not yet implemented in EMAC." This is a bit confusing, as it implies that the reactions in JAMOC are already in MECCA and some, but not all, are already in EMAC. Howeverm it does not say which ones, so it is difficult to understand how much of a change has been made. This also suggests that JAMOC is meant to replace the current aqueous-phase mechanism in MECCA and/or EMAC. As per my previous point, the relationship between the different parts of the system needs to be clarified.

   The discrepancy can be explained by the different release cycles of CAABA and EMAC. We plan to release the CAABA/MECCA box model version 4.5.0 together with this GMD paper. For our global model studies, the new chemistry was inserted into the current EMAC version 2.54. It will be available to the global modelling community when EMAC 2.55 is released, most likely in early 2021. For clarification, we removed "and reactions included in MECCA (e.g. Fenton chemistry), which are not yet implemented in EMAC" from the manuscript.

The other major comment I have is about CLEPS. The authors describe JAMOC as derived from CLEPS but they don't say how this was done. Were the reactions "hand-picked" from CLEPS (if so on which basis?) or was some reduction procedure applied?It is important that the process is described and the rationale behind certain choices is explained.

We fully agree with the reviewer that this is an important part, which is currently missing in the description of JAMOC. Therefore, we majorly revised the introduction to Sect. 2. The new version now includes a clear description of the reduction principles applied and their reasoning.

I also think it would be more useful and, more informative perhaps, to compare the output of "CAABA with JAMOC" to the output of CLEPS, rather than to the output of "CAABA without JAMOC" (Figure 3 and related discussion). This would allow a better understanding of the accuracy of the reduction procedure, and how much information (if any) is lost when the more explicit mechanism CLEPS is condensed into the smaller mechanism JAMOC.

We agree that a box model study comparison between CLEPS in its original form and JAMOC would be ideal. However, we think that a direct comparison is not feasible due to the following two issues. Even though JAMOC is derived from CLEPS, we additionally expand it to include e.g. the oligomerisation of glyoxal and the gas-phase oxidation of gem-diols. Secondly, in its original version, CLEPS is coupled to the gas-phase mechanism MCM, whereas JAMOC is coupled to MOM. Thus, a direct comparison would be highly influenced by the different gas-phase chemistry (a short comparison between MCM and MOM is presented in Sander et al., 2019).

**Minor Comments**

line 48: "only a selection of species containing up to four carbon atoms react within the aqueous-phase". Can you explain why this choice was made? I understand one of the reasons is to keep the size of the mechanism relatively small, but why only up to C4 species react and up to C10 species undergo phase transfer? Is the reaction of molecules with high carbon number too slow to matter? Do you actually need to transfer C5-C10 into the aqueous-phase if they don't react and you need to keep the mechanism small?

In CLEPS, on which JAMOC is based, only the oxidation of species containing up to four carbon atoms is represented. In the revised manuscript, we added an elaborate explanation on the selection of species taken into account (see earlier reply to a comment of the same reviewer). In the future, we plan to expand JAMOC to also include the oxidation of species with more than four carbon atoms. The uptake of species containing more than four carbon atoms is still needed to represent their removal by wet deposition in the global model EMAC (for which JAMOC is developed).

line 71: can you clarify the difference between apparent and intrinsic Henry's law constant?

We used the term "apparent Henry's law constant" as a synonym for "effective Henry's law constant". As the latter term seems to be more commonly used, we have changed our text accordingly.

line 86: "Pseudo-first order rate constants for the hydration and dehydration are mainly obtained from the literature". This implies that some were obtained or estimated in another way, please clarify.

For some species, the pseudo-first order rate constants are not known from literature. In these cases, the rates are assumed to be the same as for similar species. In the revised manuscript, this now reads as: "Pseudo-first order rate constants for the hydration and dehydration are mainly obtained from the literature (e.g. Doussin and Monod, 2013). In the case of formyldioxidanyl and hydroperoxyacetaldehyde, the pseudo-first order rate constants are assumed to be the same as for formaldehyde and glycolaldehyde, respectively."

line 141: "In all cases, branching ratios are rescaled to 100%." Can you explain this point better?

From literature, not all products are necessarily known. In order to preserve mass within the model, the branching ratios of the known products are rescaled to 100%. In the revised manuscript, this part now reads: "If products are unknown from literature, branching ratios of the known products are rescaled to 100 % in order to preserve mass."

line 149: how much faster is R11 with respect to R12?

The $HO_2$ elimination (R11) generally occurs at a rate of about $1 \times 10^6 \ s^{-1}$, whereas the $O_2^-$ elimination via $OH^-$ (R12) occurs at $4 \times 10^2 \ s^{-1}$. Here, we assume a typical $OH^-$ concentration within cloud droplets of $1 \times 10^{-7} \ M$.

line 182: correct "modells"

Done.

figure 3: the first panel should be sum of OVOC rather than sum of VOCs

Thank you for spotting this typo. We adjusted it accordingly.

**References**

Doussin, J.-F. and Monod, A.: Structure–activity relationship for the estimation of OH-oxidation rate constants of carbonyl compounds in the aqueous phase, Atmospheric Chemistry and Physics, 13, 11 625–11 641, https://doi.org/10.5194/acp-13-11625-2013, 2013.

Jöckel, P., Tost, H., Pozzer, A., Kunze, M., Kirner, O., Brenninkmeijer, C. A. M., Brinkop, S., Cai, D. S., Dyroff, C., Eckstein, J., Frank, F., Garny, H., Gottschaldt, K.-D., Graf, P., Grewe, V., Kerkweg, A., Kern, B., Matthes, S., Mertens, M., Meul, S., Neumaier, M., Nützel, M., Oberländer-Hayn, S., Ruhnke, R., Runde, T., Sander, R., Scharffe, D., and Zahn, A.: Earth System Chemistry integrated Modelling (ESCiMo) with the Modular Earth Submodel System (MESSy) version 2.51, Geoscientific Model Development, 9, 1153–1200, https://doi.org/10.5194/gmd-9-1153-2016, 2016.

Rosanka, S., Sander, R., Franco, B., Wespes, C., Wahner, A., and Taraborrelli, D.: Oxidation of low-molecular weight organic compounds in cloud droplets: global impact on tropospheric oxidants, Atmos. Chem. Phys. Discuss., https://doi.org/10.5194/acp-2020-1041, 2020.

Sander, R., Baumgaertner, A., Cabrera-Perez, D., Frank, F., Gromov, S., Grooß, J.-U., Harder, H., Huijnen, V., Jöckel, P., Karydis, V. A., Niemeyer, K. E., Pozzer, A., Riede, H., Schultz, M. G., Taraborrelli, D., and Tauer, S.: The community atmospheric chemistry box model CAABA/MECCA-4.0, Geoscientific Model Development, 12, 1365–1385, https://doi.org/10.5194/gmd-12-1365-2019, 2019.

Tost, H., Jöckel, P., Kerkweg, A., Pozzer, A., Sander, R., and Lelieveld, J.: Global cloud and precipitation chemistry and wet deposition: tropospheric model simulations with ECHAM5/MESSy1, Atmospheric Chemistry and Physics, 7, 2733–2757, https://doi.org/10.5194/acp-7-2733-2007, 2007.

---

## Author Response (AR2)

**Replies to anonymous referee #1**

The authors have satisfactorily addressed most of my comments. I only have a few minor additional remarks that could be addressed in a revised manuscript. Line numbers refer to the new manuscript without annotations.

Thank you very much for your additional review. It is good to hear that our adjustments to the manuscript are satisfactory. Please find in black your original comment and in red our reply.

l. 30: "Such a SOA formation could have a significant influence on tropospheric HOx chemistry and NO2 photolysis, which in turn affect O3."

Comment: The meaning of this sentence is not clear. Are you suggesting that due to the increased SOA burden, photolysis rates in the atmosphere are influenced? As mentioned in my previous report, the total OH (and HO2) budgets are likely not largely influenced by the reactions in the aqueous phase. Just the presence of clouds suppresses the formation of OH in the gas phase.

SOA is known to influence the aerosol optical depth and thus NO2 photolysis (Tie et al., 2005). Additionally, SOA may act as cloud condensation nuclei (Andreae and Rosenfeld, 2008). An increase in the SOA burden would thus lead to changes in cloud cover, which in return would further influence the photolysis of NO2. In our companion paper (Rosanka et al., 2020), we find that when using JAMOC, the global tropospheric HOx budget is reduced by about 7 %. Following your comment, we changed this sentence to: "By scattering, SOA is known to influence the aerosol optical depth (AOD), leading to a reduction in NO2 photolysis (Tie et al., 2005). In addition, SOA may act as a cloud condensation nuclei (CCN) (Andreae and Rosenfeld, 2008) affecting cloud properties. An increased formation of SOA would thus influence tropospheric HOx and O3 chemistry."

l. 116: "The dehydration of many gem-diols is slower than the typical lifetime of cloud droplets "

Comment: The typical lifetime of cloud droplets is on the order of several minutes. However, in the cited literature Doussin and Monod (2013), dehydration constants on the order of k(dehydr) = 0.0015 – 0.025 s⁻1 are listed which yields a half life of gem diols of  30 - 460 s (= ln 2 / k(dehydr)). Thus, I do not think that it can be generalized that gem diols do not dehydrate in cloud droplets.

We agree that the typical lifetime of a cloud droplet is in the order of several minutes. However, their typical evaporation timescale is less than 100 s (Jarecka et al., 2013). Thus, the dehydration of some gem-diols is shorter than the typical evaporation timescale of warm cloud droplets. We understand that our original formulation might be misleading and further elaboration is needed. In addition, the use of "many" might be not appropriate. Thus, we changed this formulation to: "The typical lifetime of a warm cloud droplet can be several minutes but their typical evaporation timescale is less than 100 s (Jarecka et al., 2013). Following the dehydration constants presented by Doussin and Monod (2013), the dehydration of some gem-diols can be slower than the typical cloud droplet evaporation timescale. Additionally, their rapid transfer across the phases is expected to affect the gas-phase concentration of gem-diols, for which no other significant source is known. This process could be an important removal of gem-diols from the aqueous-phase, without yielding the original aldehyde. Therefore, their outgassing is considered for use with the models representing evaporating clouds like the EMAC model (following Sect. 2.2)."

l. 160: "The formation of oligomers within cloud droplets is known to be a source of in-cloud SOA formation"

Comment: I think that the authors misunderstand the difference between oligomers and cloud SOA. I agree that SOA formation in clouds may be an efficient SOA source. However, this SOA is mostly not composed of oligomers. Tan et al. (2009) and others discussed that SOA formation in clouds does not lead to oligomers but can be largely explained by the oxidation of small organics (e.g. glyoxal) by OH leading to oxalic acid and other carboxylic acids. While in the study by Lin et al. (2012) the same reactive uptake coefficients for aerosol and clouds were used (in accordance with an earlier global model study), these authors also state "Furthermore, the same uptake coefficient for both cloud droplets and aqueous aerosols cannot account for the differences in the chemistry of carbonyl compounds between cloud water and aerosol water (Lim et al., 2010; Ervens and Volkamer, 2010)." Thus, they are aware that different chemical pathways exist in cloud and aerosol but could

not account for it back then due to the lack of appropriate reaction parameters.

Therefore, I suggest to either remove the introductory sentence of this section or to reword it to "The formation of oligomers within the atmospheric aqueous phase is known to be a source of SOA."

Thank you very much for the clarification. We agree that your formulation is more suited. Therefore, we changed it in the revised manuscript.

l. 202: "In general, the photolysis of organic compounds competes with the other oxidation pathways (see Sect. 2.5) and is a major source of OH. In Rosanka et al. (2020a), the photolysis of OVOCs is estimated to be more than four time higher than the photolysis of H2O2."

Comment: 1) This reads as if all organics can act as OH source. Please specify that only organic peroxides are photolysed.

We agree with the referee that our formulation could imply that all organics photolyse. However, only specifying "organic peroxides" here could be misleading because also non organic peroxides undergo photolysis (e.g. pyruvic acid) in JAMOC. Therefore, we changed it to: "In general, the photolysis of some organic compounds (e.g. organic peroxides, pyruvic acid) competes with other oxidation pathways (see Sect. 2.5) and can be a source of OH."

Comment: 2) Please add a reference to an experimental study that demonstrates the much higher photolysis rate of organics as compared to H2O2.

Experimental studies reporting aqueous-phase photolysis rates are rare, and we are not aware of any experimental study performing a direct comparison of organic photolysis rates to H2O2. In addition, it is important to notice that some of the photolysis rates used in JAMOC are taken from the gas-phase and scaled using an enhancement factor to account for scattering effects. Thus, in the model the relative magnitude between J(H2O2) and J(organics) is mainly the one based on the experimental data for the gas phase. The values reported in Rosanka et al. (2020) are modelling results that provide a global tropospheric in-cloud OH budget and a comparison to experimental data is thus limited and hardly possible. Following your comment, we propose to rewrite the sentence to the following: "In Rosanka et al. (2020a), a global tropospheric in-cloud OH budget is presented. When using JAMOC, EMAC predicts that about 40 % of all in-cloud OH is produced from the photolysis of a selection of organic compounds. However, Fenton chemistry is not considered by Rosanka et al. (2020) and the relative contribution is therefore expected to be overestimated."

Comment: 3) Typo: it should be 'times' not 'time' in the last sentence.

Following our suggestions to your "Comment: 2)", this typo no longer exists.

**References**

Andreae, M. and Rosenfeld, D.: Aerosol–cloud–precipitation interactions. Part 1. The nature and sources of cloud-active aerosols, Earth-Science Reviews, 89, 13–41, https://doi.org/https://doi.org/10.1016/j.earscirev.2008.03.001, 2008.

Doussin, J.-F. and Monod, A.: Structure–activity relationship for the estimation of OH-oxidation rate constants of carbonyl compounds in the aqueous phase, Atmospheric Chemistry and Physics, 13, 11 625–11 641, https://doi.org/10.5194/acp-13-11625-2013, 2013.

Ervens, B. and Volkamer, R.: Glyoxal processing by aerosol multiphase chemistry: towards a kinetic modeling framework of secondary organic aerosol formation in aqueous particles, Atmospheric Chemistry and Physics, 10, 8219–8244, https://doi.org/10.5194/acp-10-8219-2010, 2010.

Jarecka, D., Grabowski, W. W., Morrison, H., and Pawlowska, H.: Homogeneity of the Subgrid-Scale Turbulent Mixing in Large-Eddy Simulation of Shallow Convection, Journal of the Atmospheric Sciences, 70, 2751 – 2767, https://doi.org/10.1175/JAS-D-13-042.1, 2013.

Lim, Y. B., Tan, Y., Perri, M. J., Seitzinger, S. P., and Turpin, B. J.: Aqueous chemistry and its role in secondary organic aerosol (SOA) formation, Atmospheric

Chemistry and Physics, 10, 10521–10539, https://doi.org/10.5194/acp-10-10521-2010, URL https://acp.copernicus.org/articles/10/10521/2010/, 2010.

Lin, G., Penner, J. E., Sillman, S., Taraborrelli, D., and Lelieveld, J.: Global modeling of SOA formation from dicarbonyls, epoxides, organic nitrates and peroxides, Atmospheric Chemistry and Physics, 12, 4743–4774, https://doi.org/10.5194/acp-12-4743-2012, 2012.

Rosanka, S., Sander, R., Franco, B., Wespes, C., Wahner, A., and Taraborrelli, D.: Oxidation of low-molecular weight organic compounds in cloud droplets: global impact on tropospheric oxidants, Atmos. Chem. Phys. Discuss., https://doi.org/10.5194/acp-2020-1041, 2020.

Tan, Y., Perri, M. J., Seitzinger, S. P., and Turpin, B. J.: Effects of Precursor Concentration and Acidic Sulfate in Aqueous Glyoxal–OH Radical Oxidation and Implications for Secondary Organic Aerosol, Environmental Science & Technology, 43, 8105–8112, https://doi.org/10.1021/es901742f, pMID: 19924930, 2009.

Tie, X., Madronich, S., Walters, S., Edwards, D. P., Ginoux, P., Mahowald, N., Zhang, R., Lou, C., and Brasseur, G.: Assessment of the global impact of aerosols on tropospheric oxidants, Journal of Geophysical Research: Atmospheres, 110, https://doi.org/https://doi.org/10.1029/2004JD005359, 2005.

---

## Author Response (AR3)

Dear Christoph Knote,

thank you very much for accepting our manuscript. Following you request, we performed the following technical modifications to the manuscript:

1. The version number was updated from '4.5.0gmdd' to '4.5.0'
2. We archived the model code at Zenodo (http://doi.org/10.5281/zenodo.4707938)
3. We archived the model output at Jülich DATA (https://doi.org/10.26165/JUELICH-DATA/SD9F6B)
4. The model runtimes reported in section 4 were updated to the values obtained with the archived version of the model code

At this point, we would like to thank you for serving as an editor of this manuscript.

Kind regards,
On behalf of the authors,

Simon Rosanka